# Contingency and chance erase necessity in the experimental evolution of ancestral proteins

Victoria Cochran Xie[1][†], Jinyue Pu[1][†]*, Brian PH Metzger[2][†], Joseph W Thornton[2,3]*, Bryan C Dickinson[1]*

[1]Department of Chemistry, University of Chicago, Chicago, United States; [2]Department of Ecology and Evolution, University of Chicago, Chicago, United States; [3]Department of Human Genetics, University of Chicago, Chicago, United States

**Abstract** The roles of chance, contingency, and necessity in evolution are unresolved because they have never been assessed in a single system or on timescales relevant to historical evolution. We combined ancestral protein reconstruction and a new continuous evolution technology to mutate and select proteins in the B-cell lymphoma-2 (BCL-2) family to acquire protein–protein interaction specificities that occurred during animal evolution. By replicating evolutionary trajectories from multiple ancestral proteins, we found that contingency generated over long historical timescales steadily erased necessity and overwhelmed chance as the primary cause of acquired sequence variation; trajectories launched from phylogenetically distant proteins yielded virtually no common mutations, even under strong and identical selection pressures. Chance arose because many sets of mutations could alter specificity at any timepoint; contingency arose because historical substitutions changed these sets. Our results suggest that patterns of variation in BCL-2 sequences – and likely other proteins, too – are idiosyncratic products of a particular and unpredictable course of historical events.

**\*For correspondence:**
pujy@uchicago.edu (JP);
joet1@uchicago.edu (JWT);
Dickinson@uchicago.edu (BCD)

[†]These authors contributed equally to this work

## Introduction

The extent to which biological diversity is the necessary result of optimization by natural selection or the unpredictable product of random events and historical contingency is one of evolutionary biology's most fundamental and unresolved questions (*Gould, 1989*; *Jablonski, 2017*; *Ramsey and Pence, 2016*; *Travisano et al., 1995*). The answer would have strong implications not only for our understanding of evolutionary processes but also for how we should analyze the particular forms of variation that exist today. For example, if diversity primarily reflects a predictable process of adaptation to distinct environments, then a central goal of biology would be to explain how the characteristics of living things help to execute particular functions and improve fitness (*Mayr, 1983*). By contrast, if diversity reflects chance sampling from a set of similarly fit possibilities, then the variation itself is of little interest because it does not affect biological properties or shape future evolutionary outcomes; the goal of biology would be to identify the invariant characteristics of natural systems and explain how they contribute to function (*Kimura, 1983*; *Lobkovsky and Koonin, 2012*; *Monod, 1972*; *Morris, 2015*). Finally, if diversity reflects contingency – a strong dependence of future outcomes on initial conditions or subsequent events, also known as path-dependence – then the outcomes of evolution would be predictable only given complete knowledge of the constraints and opportunities specific to each set of conditions (*Beatty, 2009*; *Blount et al., 2018*; *Desjardins, 2011*; *Gould and Lewontin, 1979*); the goal of biology would then be to characterize these constraints and opportunities, their mechanistic causes, and the historical events that shaped them.

**eLife digest** One of the most fundamental and unresolved questions in evolutionary biology is whether the outcomes of evolution are predictable. Is the diversity of life we see today the expected result of organisms adapting to their environment throughout history (also known as natural selection) or the product of random chance? Or did chance events early in history shape the paths that evolution could take next, determining the biological forms that emerged under natural selection much later?

These questions are hard to study because evolution happened only once, long ago. To overcome this barrier, Xie, Pu, Metzger et al. developed an experimental approach that can evolve reconstructed ancestral proteins that existed deep in the past. Using this method, it is possible to replay evolution multiple times, from various historical starting points, under conditions similar to those that existed long ago. The end products of the evolutionary trajectories can then be compared to determine how predictable evolution actually is.

Xie, Pu, Metzger et al. studied proteins belonging to the BCL-2 family, which originated some 800 million years ago. These proteins have diversified greatly over time in both their genetic sequences and their ability to bind to specific partner proteins called co-regulators. Xie, Pu, Metzger et al. synthesized BCL-2 proteins that existed at various times in the past. Each ancestral protein was then allowed to evolve repeatedly under natural selection to acquire the same co-regulator binding functions that evolved during history.

At the end of each evolutionary trajectory, the genetic sequence of the resulting BCL-2 proteins was recorded. This revealed that the outcomes of evolution were almost completely unpredictable: trajectories initiated from the same ancestral protein produced proteins with very different sequences, and proteins launched from different ancestral starting points were even more dissimilar.

Further experiments identified the mutations in each trajectory that caused changes in coregulator binding. When these mutations were introduced into other ancestral proteins, they did not yield the same change in function. This suggests that early chance events influenced each protein's evolution in an unpredictable way by opening and closing the paths available to it in the future.

This research expands our understanding of evolution on a molecular level whilst providing a new experimental approach for studying evolutionary drivers in more detail. The results suggest that BCL-2 proteins, in all their various forms, are unique products of a particular, unpredictable course of history set in motion by ancient chance events.

Many studies have provided insight into the ways that chance, contingency, and necessity can affect the evolution of molecular sequences and functions, but the relative importance of these factors during evolutionary history remains unresolved because they have never been measured in the same system, and their effects over long evolutionary time scales have not been characterized. For example, experiments on ancestral proteins have shown that particular historical mutations have different effects when introduced into different ancestral backgrounds – suggesting contingency – but they do not reveal the extent to which context-dependence actually influenced evolutionary outcomes; further, these historical trajectories happened only once, so they cannot elucidate the effect of contingency relative to chance (*Bloom et al., 2010*; *Bridgham et al., 2009*; *Gong et al., 2013*; *Harms and Thornton, 2014*; *McKeown et al., 2014*; *Natarajan et al., 2016*; *Ortlund et al., 2007*; *Risso et al., 2015*; *Starr et al., 2018*; *Wu et al., 2018*). Experimental evolution studies could, in principle, characterize both chance and contingency if they had sufficient replication from multiple starting points, but to date no study has done so; furthermore, no study has imposed selection on historical proteins to acquire functions that changed during history, so their relevance to historical evolution is not clear (*Baier et al., 2019*; *Blount et al., 2012*; *Bollback and Huelsenbeck, 2009*; *Couñago et al., 2006*; *Dickinson et al., 2013*; *Kacar et al., 2017*; *Kryazhimskiy et al., 2014*; *Meyer et al., 2012*; *Salverda et al., 2011*; *Spor et al., 2014*; *van Ditmarsch et al., 2013*; *Wichman et al., 1999*; *Wünsche et al., 2017*; *Zheng et al., 2019*). Studies of phenotypic convergence in nature suggest some degree of repeatability at the genetic level (reviewed in *Arendt and Reznick, 2008*; *Gompel and Prud'homme, 2009*; *Orgogozo, 2015*; *Storz, 2016*), but these studies

rarely involve replicate lineages from the same starting genotypes, and evolutionary conditions are seldom identical; as a result, similarities and differences among lineages cannot be attributed to chance, contingency, or necessity. Furthermore, these studies have typically involved closely related species or populations and therefore do not measure the effects of chance and contingency that might be generated during long-term evolution.

The ideal experiment to determine the relative roles of chance, contingency, and necessity in historical evolution would be to travel back in time, re-launch evolution multiple times from each of various starting points that existed during history, and allow these trajectories to play out under historical environmental conditions (*Gould, 1989*). By comparing outcomes among replicates launched from the same starting point, we could estimate the effects of chance; by comparing those from different starting points, we could quantify the effects of contingency that was generated along historical evolutionary paths (*Figure 1*). Necessity would be apparent if the same outcome recurred in every replicate, irrespective of the point from which evolutionary trajectories were launched and changes that occurred subsequently: in that case, evolution would be both deterministic (free of chance) and insensitive to initial and intervening conditions (noncontingent). Although time travel is currently impossible, we can approximate this ideal design by reconstructing ancestral proteins as they existed in the deep past (*Thornton, 2004*) and using them to launch replicated evolutionary trajectories in the laboratory under selection to acquire the same molecular functions that evolved during history.

Here we implement this strategy using the B-cell lymphoma-2 (BCL-2) protein family as a model system and the specificity of protein–protein interactions (PPIs) as the target of selection. BCL-2 family proteins are involved in the regulation of apoptosis (*Chipuk et al., 2010*; *Danial and Korsmeyer, 2004*; *Kale et al., 2018*; *Petros et al., 2004*) through PPIs with coregulators (*Chen et al., 2005*; *Chen et al., 2013*; *Dutta et al., 2010*; *Lomonosova and Chinnadurai, 2008*). Although there are many dimensions to BCL-2 family proteins' cellular effects, different binding specificities for coregulator proteins are a critical determinant of their particular biological functions. Among BCL-2 family members, the myeloid cell leukemia sequence 1 protein (MCL-1) class strongly binds both the BID and NOXA coregulators, whereas the BCL-2 class (a subset of the larger BCL-2 protein family) strongly binds BID but not NOXA (*Figure 2A*; *Certo et al., 2006*). The two classes share an ancient evolutionary origin: both are found throughout the Metazoa (*Banjara et al., 2020*; *Lanave et al., 2004*) and are structurally similar, using the same cleft to interact with their coregulators (*Figure 2B*, *Figure 2—figure supplement 1*), despite having only 20% sequence identity.

To drive the evolution of new PPI specificities, we developed a new high-throughput phage-assisted continuous evolution (PACE) system (*Esvelt et al., 2011*) that can simultaneously select for and against particular PPIs (*Pu et al., 2019*; *Pu et al., 2017b*). We applied this technique to a series of reconstructed ancestral BCL-2 family members, repeatedly evolving each starting genotype to acquire PPI specificities found among extant family members. By comparing sequence outcomes among PACE replicates from the same starting point, we quantified the role of chance in the evolution of historically relevant molecular functions under strong and identical selection pressures; by comparing outcomes of PACE initiated from different starting points, we quantified the effect of contingency generated by the sequence changes that accumulated during these proteins' histories. This design also allowed us to characterize how these factors have changed over phylogenetic time and dissect the underlying genetic basis by which they emerged.

## Results

### BID specificity is derived from an ancestor that bound both BID and NOXA

We first characterized the historical evolution of PPI specificity in the BCL-2 family using ancestral protein reconstruction (*Figure 2—figure supplement 2*). We inferred the maximum likelihood phylogeny of the family, which recovered the expected sister relationship between the metazoan BCL-2 and MCL-1 classes (*Figure 2C*, *Figure 2—figure supplement 3*). We then reconstructed the most recent common ancestor (AncMB1) of the two classes – a gene duplication that occurred before the last common ancestor (LCA) of all animals – and 11 other ancestral proteins that existed along the

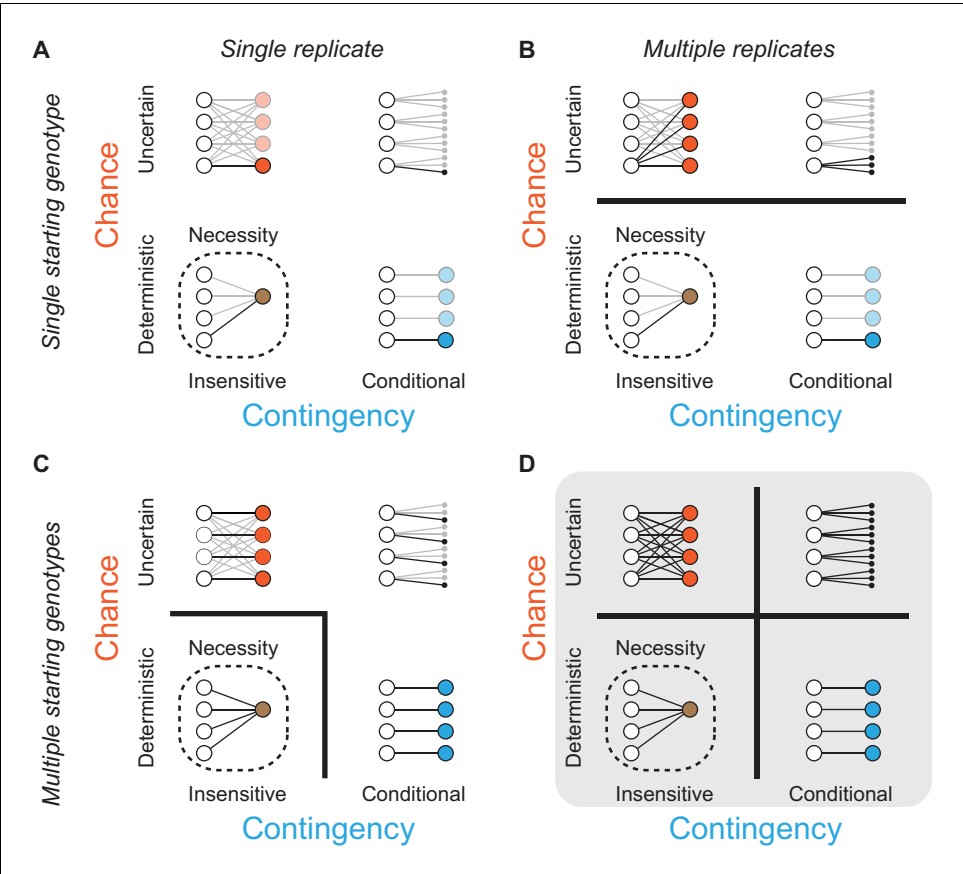

**Figure 1.** Assessing the effects of chance and contingency during evolution. Each panel (A-D) shows the capacity of one experimental design to detect chance and contingency; the quadrants within each panel show evolutionary scenarios with varying degrees of chance and contingency. Chance (y-axis within each panel) is defined as random occurrence of events from a probability distribution in which multiple events have probability > 0 given some defined starting point; in the absence of chance, evolution is deterministic because a single outcome always occurs from any starting genotype. Contingency (x-axis within each panel) is defined as differences in this probability distribution given different starting or subsequent conditions; in the absence of contingency, outcomes are insensitive to these conditions, and all starting points lead to the same outcome or set of outcomes. Lines connect starting genotypes (white circles) to evolutionary outcomes. Quadrants show evolution under the influence of chance (orange), contingency (blue), or both (black); outcomes are necessary (brown, with dotted line) when neither chance nor contingency is important. Potential trajectories that are not observed because of deficiencies in experimental design are shown with reduced opacity. Thick black lines between quadrants in (A–D) separate evolutionary scenarios that can be distinguished from each other given each design. (A) Assessing one evolutionary replicate from one starting point provides no information about the extent to which chance, contingency, or necessity shape the outcome. (B) Assessing multiple replicates from one starting point can detect chance but provides no information about contingency. (C) Assessing one replicate each from multiple starting points can detect necessity or its absence, but cannot not distinguish between chance and contingency. (D) Studying multiple replicates from multiple starting genotypes allows chance, contingency, and necessity to be distinguished.

lineages leading from AncMB1 to human BCL-2 (hsBCL-2) and to human MCL-1 (hsMCL-1) (*Supplementary file 1*).

We synthesized genes coding for these proteins and experimentally assayed their ability to bind BID and NOXA using a proximity-dependent split RNA polymerase (RNAP) luciferase assay (*Figure 2—figure supplement 4*; *Pu et al., 2017b*). AncMB1 bound both BID and NOXA, as did all ancestral proteins in the MCL-1 clade and hsMCL-1 (*Figure 2C*, *Supplementary file 1*). Ancestral proteins in the BCL-2 clade that existed before the LCA of deuterostomes also bound both BID and NOXA, whereas BCL-2 ancestors within the deuterostomes bound only BID, just as hsBCL-2 does.

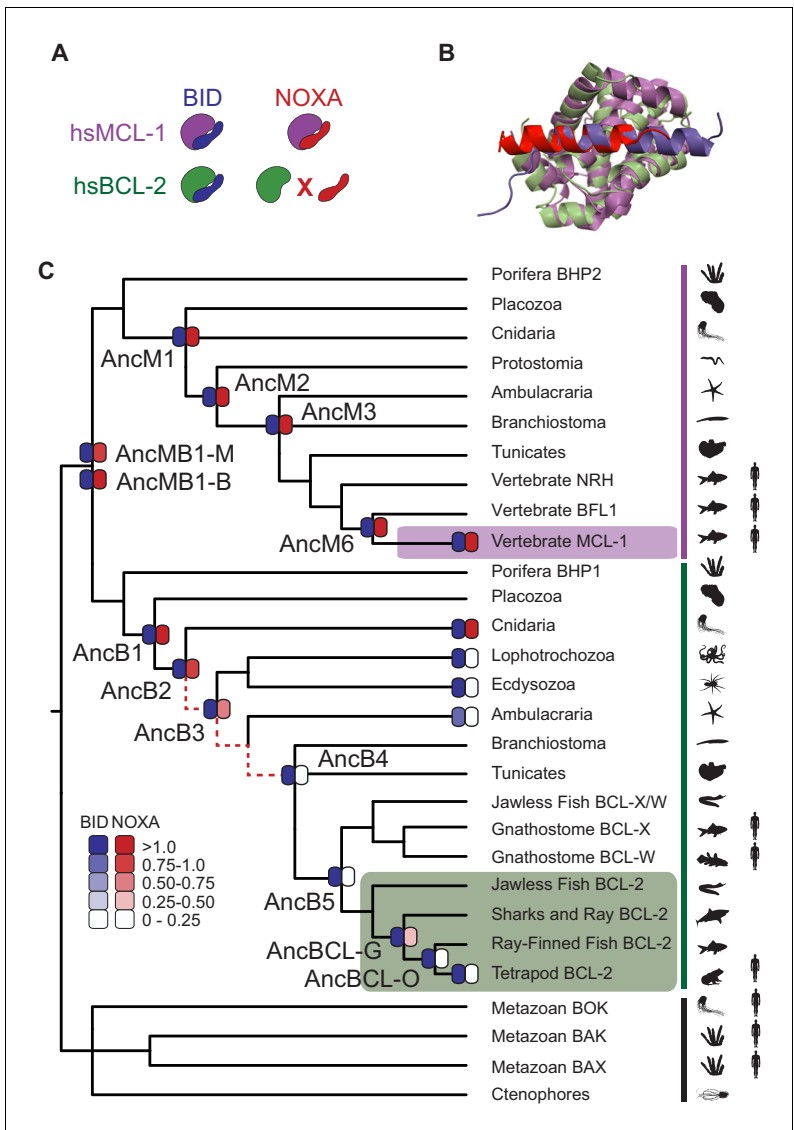

**Figure 2.** BID specificity was acquired during vertebrate BCL-2 evolution. (**A**) Protein binding specificities of extant BCL-2 family members. Human MCL-1 (hsMCL-1, purple) strongly binds BID (blue) and NOXA (red), while human BCL-2 (hsBCL-2, green) strongly binds BID but not NOXA. (**B**) Crystal structures of MCL-1 (purple) bound to NOXA (red, PDB 2nla), and BCL-xL (green, a closely related paralog of BCL-2) bound to BID (blue, PDB 4qve). (**C**) Reduced maximum likelihood phylogeny of BCL-2 family proteins. Purple bar, MCL-1 class; green bar, BCL-2 class. The phylogeny was rooted using as outgroups the paralogs BOX, BAK, and BAX (black bar). Heatmaps indicate BID (blue) and NOXA (red) binding measured using the luciferase assay. Each shaded box shows the normalized mean of three biological replicates. Red dotted lines, interval during which NOXA binding was lost, yielding BID specificity in the BCL-2 proteins of vertebrates (green box). Purple box, vertebrate MCL-1. Silhouettes, representative species in each terminal group. AncMB1-M and -B are alternative reconstructions using different approaches to alignment ambiguity (see Materials and methods). For complete phylogeny, see *Figure 2—figure supplement 3*.

The online version of this article includes the following figure supplement(s) for figure 2:

**Figure supplement 1.** BCL-2 family proteins are structurally similar but have different binding profiles.

**Figure supplement 2.** Ancestral sequence reconstruction procedure in schematic form.

**Figure supplement 3.** Maximum likelihood phylogeny of BCL-2 family proteins.

**Figure supplement 4.** Binding of BID and NOXA to extant and ancestral proteins.

This reconstruction of history was robust to uncertainty in the ancestral sequences: experiments on 'AltAll' proteins at each ancestral node – which combine all plausible alternative amino acid states (posterior probability > 0.2) in a single 'worst-case' alternative reconstruction – also showed that BID specificity arose within the BCL-2 clade (*Figure 2—figure supplement 4*, *Supplementary file 2*).

To further test this inferred history, we characterized the coregulator specificity of extant BCL-2 class proteins from taxonomic groups in particularly informative phylogenetic positions. Those from Cnidaria were activated by both BID and NOXA, whereas those from protostomes and invertebrate deuterostomes were BID-specific (*Figure 2C*, *Figure 2—figure supplement 4*, *Supplementary file 1*). These results corroborate the inferences made from ancestral proteins, indicating that BID specificity evolved when the ancestral ability to bind NOXA was lost between AncB2 (in the ancestral eumetazoan) and AncB4 (in the ancestral deuterostome).

## A directed continuous evolution system for rapid changes in PPI specificity

To rapidly evolve BCL-2 family proteins to acquire the same PPI specificities that existed during the family's history, we developed a new PACE system (*Esvelt et al., 2011*; *Figure 3A–B*, *Figure 3—figure supplement 1*). Previous PACE systems have evolved binding to new protein partners using a bacterial 2-hybrid approach (*Badran et al., 2016*), but evolving PPI specificity requires simultaneous selection for a desired PPI and against an undesired PPI. For this purpose, we used two orthogonal proximity-dependent split RNAPs that recognize different promoters in the same cell and – if reconstituted by a PPI – activate transcription of positive and negative selectable markers. Specifically, the N-terminal fragment of RNAP was fused to the BCL-2 protein of interest and encoded in the phage genome, and two C-terminal RNAP fragments (RNAPc), each fused to a different BCL-2 coregulator, were encoded on host cell plasmids. One RNAPc is fused to the selected-for coregulator and drives expression of an essential viral gene (gIII) when reconstituted by binding to the BCL-2 protein; the other RNAPc, fused to the counter-selected coregulator, drives expression of a dominant-negative version of gIII (*Pu et al., 2017a*). Phage containing BCL-2 variants that bind the positive selection protein but not the counterselection protein produce infectious phage. After optimizing this system, we used activity-dependent plaque assays and phage growth assays to confirm that it imposes strong selection for the PPI specificity profiles of extant hsBCL-2 and hsMCL1 (*Figure 3D*).

The simplicity of this platform allowed us to drive extant and reconstructed ancestral proteins to recapitulate or reverse the historical evolution of the BCL-2 family's PPI specificity in multiple replicates in just days, without severe experimental bottlenecks. Three proteins that bound both BID and NOXA – hsMCL-1, AncM6, and AncB1 – were selected to acquire the derived BCL-2 phenotype, retaining BID binding and losing NOXA binding. Conversely, hsBCL-2, AncB5, and AncB4 were evolved to gain NOXA binding, reverting to the ancestral phenotype (*Figure 3C and E*, *Figure 3—figure supplement 2*). For each starting genotype, we performed four replicate experimental evolution trajectories (*Supplementary file 3*). Each experiment was run for 4 days, corresponding to approximately 100 rounds of viral replication (*Esvelt et al., 2011*). All trajectories yielded the target PPI specificity, which we confirmed by experimental analysis of randomly isolated phage clones using activity-dependent plaque assays and in vivo and in vitro binding assays (*Figure 4A–B*, *Figure 3—figure supplement 2*, *Figure 3—figure supplement 3*). As in prior PACE experiments, variation in the selected phenotype was observed among individual phage isolates within the final populations (*Dickinson et al., 2013*), presumably because of large populations, high mutation rates, and/or inadequate time for fixation.

## Chance and contingency erase necessity in the evolution of PPI specificity

We used deep sequencing to compare the sequence outcomes of evolution across trajectories initiated from the same and different starting points (*Figure 4—figure supplement 1*). Necessity was almost entirely absent. Across all trajectories, 100 mutant amino acid states at 75 different sites evolved to frequency > 5% in at least one replicate (*Figure 4C–D*, *Figure 4—figure supplement 2*, *Supplementary file 4*). Of these acquired states, 73 appeared in only a single trajectory, and only four arose in more than one replicate from multiple starting points (*Figure 4E*, *Figure 4—figure supplement 3*). When selection was imposed for binding to both BID and NOXA, no states were

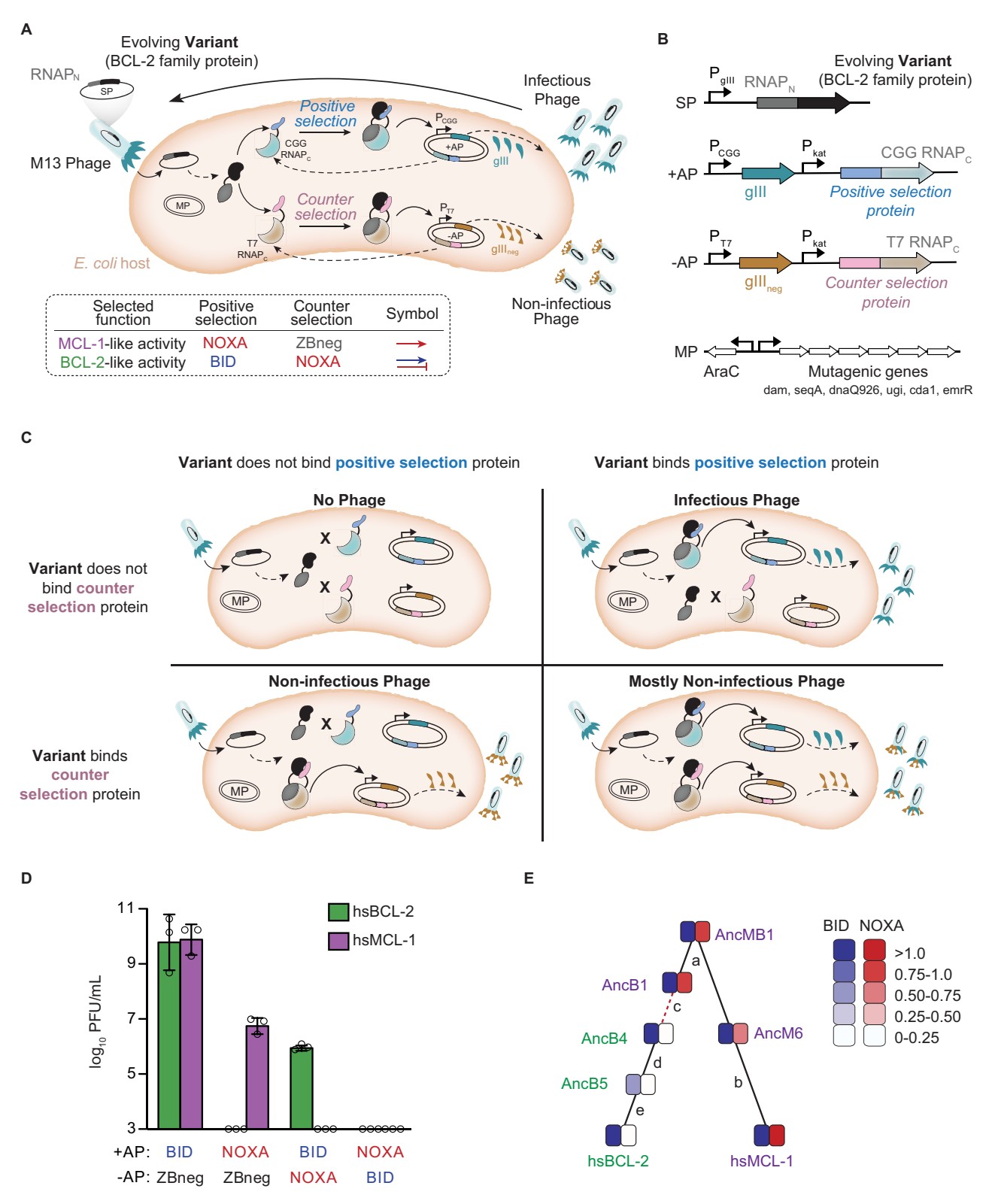

**Figure 3.** Continuous directed evolution of specificity in modern and ancestral BCL-2 family proteins. (**A**) *Top:* Components of the PACE system for evolving PPI specificity. Solid arrows show potential binding events. Dashed arrows show potential protein expression. The protein targeted for altered specificity (black) is fused to the N-terminus of RNA polymerase (RNAP_N, dark gray) and placed into the M13 phage genome (SP, selection plasmid). Upon infection of host *E. coli*, the target gene-RNAP_N fusion is expressed. Host cells carry accessory plasmids (+AP and −AP) that contain the

*Figure 3 continued on next page*

*Figure 3 continued*

C-terminus of RNAP (RNAP$_C$) fused to peptides for which specificity is desired (blue, positive selection protein; pink, counterselection protein). Binding of the target protein to either the selection protein or counterselection protein reconstitutes a functional RNAP. Binding of RNAP to the corresponding promoter results in the expression of either gIII (teal) or gIII$_{neg}$ (gold). gIII is necessary to produce infectious phage. gIII$_{neg}$ is a dominant-negative version of gIII which results in the production of non-infectious phage. An arabinose-inducible mutagenesis plasmid in the system (MP) increases the mutation rate of the evolving protein. *Bottom*: PACE schemes for evolving PPI specificities. To select for BCL-2 like specificity, positive selection to bind BID was imposed with counterselection to avoid binding NOXA (blue arrow and red bar). To evolve MCL-1 like activity, positive selection to bind NOXA (red arrow) was imposed after a phase of selection for BID binding, both with counterselection to avoid nonspecific binding using a control zipper peptide (ZBneg). (B) Map of the phage SP, the positive and counterselection accessory plasmids (+AP and −AP), and the MP. (C) Selection for protein variants with the desired specificity. *Upper left*: Infection by a phage carrying a protein variant that binds neither the positive selection nor the counterselection protein results in production of little to no progeny phage. *Upper right*: Infection by a phage carrying a protein variant that binds only the positive selection protein results in expression of gIII and production of infectious phage. *Lower left*: Infection by a phage carrying a protein variant that binds only the counterselection protein results in expression of gIII$_{neg}$ and production of non-infectious phage. *Lower right*: Infection by a phage carrying a protein variant that binds the positive selection and counterselection proteins results in expression of both gIII and gIII$_{neg}$, leading to production of primarily non-infectious phage. (D) Growth assays to assess selection and counterselection. Plaque forming units (PFU) after culturing 1000 phage-containing hsBCL-2 (green) or hsMCL-1 (purple) on *E. coli* containing various APs. Detection limit 10$^3$ PFU/mL. Bars show mean ± SD of three replicates (circles). (E) Phylogenetic relations of starting genotypes used in PACE. Each starting genotype was selected to acquire a new specificity in four independent replicates. Green, proteins selected to gain NOXA binding; purple, proteins selected to lose NOXA binding. Red dashed line, interval during which NOXA binding was historically lost, yielding BID specificity in the BCL-2 clade. Letters, index of phylogenetic intervals between ancestral proteins referred to in *Figure 5*.

The online version of this article includes the following figure supplement(s) for figure 3:

**Figure supplement 1.** Using PACE to evolve target PPI specificity of BCL-2 family proteins.
**Figure supplement 2.** Selection schemes and phage titers for changes in PPI specificity.
**Figure supplement 3.** Fluorescence polarization of PACE-evolved variants.

predictably acquired in all trajectories from all starting points. The only mutation universally acquired under any selection regime was a nonsense mutation at codon 271, which was acquired in all trajectories selected for BID specificity, but experimental analysis of this mutation shows that it has no detectable effect on coregulator binding (*Figure 4—figure supplement 4*).

Both chance and contingency contributed to this pervasive unpredictability. Pairs of trajectories launched from the same starting point differed, on average, at 78% of their acquired states, indicating a strong role for chance. Pairs that were launched from different starting points (but selected for the same PPI specificity) differed at an average of 92% of acquired states, indicating an additional role for contingency.

These starting points are separated by different amounts of evolutionary divergence, so to understand the extent of contingency over the timescale of metazoan evolution, we compared trajectories launched from AncB1 to those launched from hsMCL1 (the two most distant genotypes that were selected for BID specificity). Of 34 states acquired in these experiments, only three occurred in at least one trajectory from both starting points. Of 40 states acquired in trajectories launched from AncB4 and hsBCL-2 (the two most distant proteins that were selected to gain NOXA binding), only one occurred in any trajectories from both starting points. Together, contingency generated across long phylogenetic timescales and chance therefore make sequence evolution in the BCL-2 family almost entirely unpredictable.

These experiments indicate an almost complete lack of necessity in the evolution of PPI specificity in PACE. To gain insight into the extent of necessity in the historical evolution of BCL-2 PPI specificity, we asked whether substitutions that occurred during the phylogenetic interval when NOXA binding was lost (between AncB1 and AncB4) were either repeated or reversed during PACE trajectories to lose or regain NOXA binding from any starting point (*Figure 4F*, *Figure 4—figure supplement 5*, *Figure 4—figure supplement 6*). In PACE experiments to lose NOXA binding from proteins that initially bound both peptides, none of the acquired states recapitulated substitutions from the branch on which NOXA binding was historically lost. In PACE experiments to reacquire NOXA binding from proteins with BCL2-like specificity for BID, only two states reversed historical substitutions that occurred on that branch. Both of these reacquisitions occurred in PACE trajectories launched from AncB4, the immediate daughter node of this branch, suggesting that in other proteins, contingency accumulated over phylogenetic time restricted their accessibility. Furthermore, both of these states were acquired in only a subset of trajectories from AncB4, indicating a role for chance even

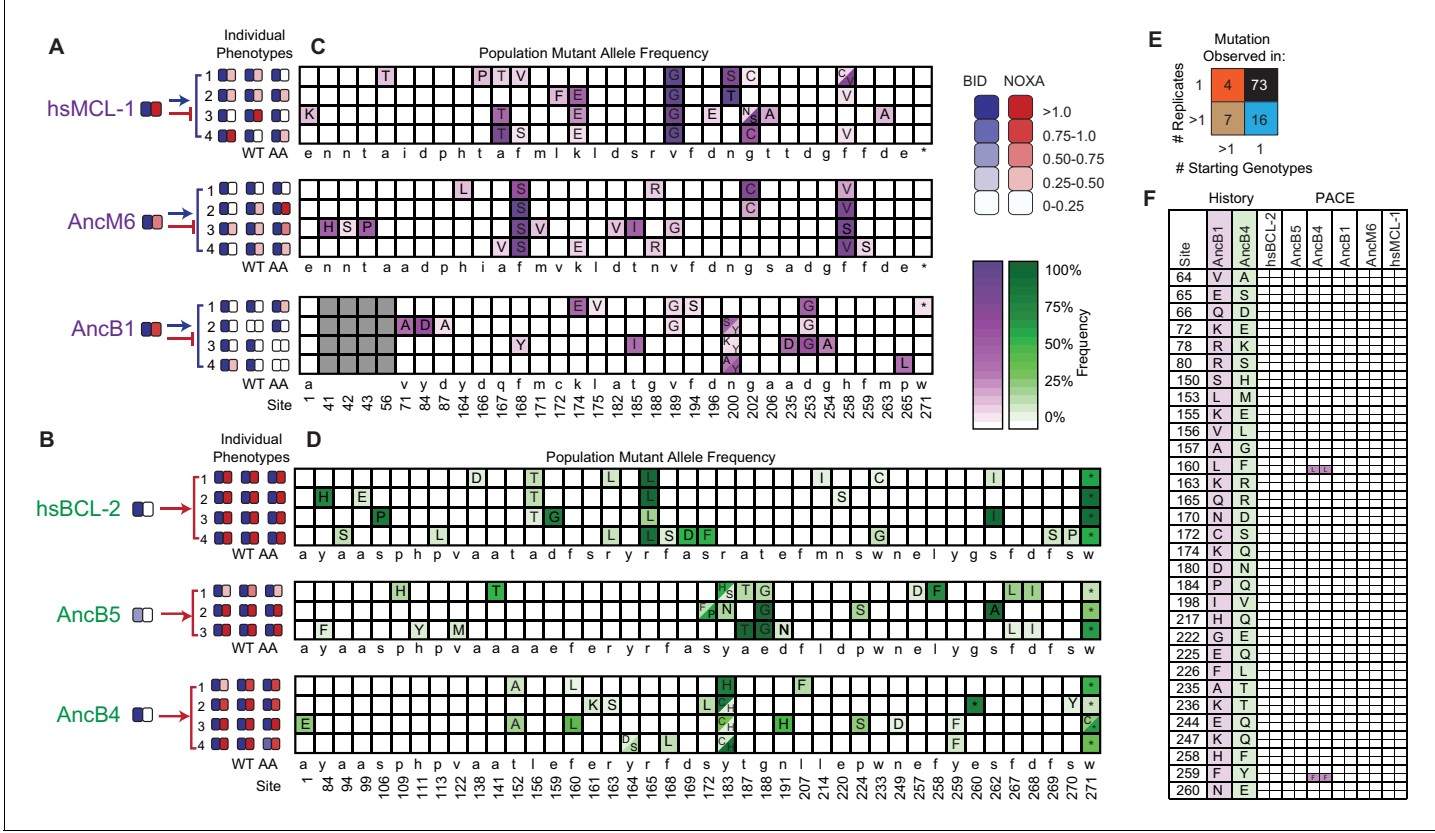

**Figure 4.** Chance and contingency shape evolutionary outcomes. (**A**) Phenotypic outcome of PACE experiments when proteins with MCL-1-like specificity were selected to maintain BID and lose NOXA binding. For each starting genotype, the BID (blue) and NOXA (red) binding activity of the starting genotype and three phage variants isolated from each evolved replicate (number) are shown as heatmaps. (**B**) Phenotypic outcome of PACE experiments when proteins with BCL-2-like specificity were selected to gain NOXA binding. (**C**) Frequency of acquired states in PACE experiments when proteins with MCL-1-like specificity were selected to maintain BID and lose NOXA binding. Rows, outcomes of each replicate trajectory. Columns, sites that acquired one or more non-wild-type amino acids (letters in cells) at frequency >5%; color saturation shows the frequency of the acquired state. Site numbers and wild-type amino acid (WT AA) states are listed. Gray, sites that do not exist in AncB1. (**D**) Frequency of acquired states when BCL-2-like proteins were selected to gain NOXA binding. (**E**) Repeatability of acquired states across replicates. The 100 non-WT states acquired in all experiments were categorized as occurring in 1 or >1 replicate trajectory from 1 or >1 unique starting genotype, with the number in each category shown. The vast majority of states evolved in just one replicate from one starting point (black). (**F**) Historical substitutions that contributed to the change in PPI specificity rarely occur or revert during PACE. Rows, substitutions that historically occurred between AncB1 and AncB4, the ancestral proteins that flank the loss of NOXA on the phylogeny. For each substitution, columns show whether the historical ancestral or derived state was acquired in PACE trajectories from each ancestral starting point. Purple and green boxes, PACE acquisition of ancestral or derived state, respectively, in each replicate. White boxes, neither state acquired.

The online version of this article includes the following figure supplement(s) for figure 4:

**Figure supplement 1.** MiSeq library preparation.
**Figure supplement 2.** Frequency of insertions and deletions during PACE.
**Figure supplement 3.** Categories of the 100 non-WT states observed for each non-WT state.
**Figure supplement 4.** Effect of w271* mutation on BID and NOXA binding.
**Figure supplement 5.** Historical distribution of PACE mutations.
**Figure supplement 6.** Phylogenetic recapitulation of PACE mutations.

from this starting point. Some substitutions that occurred during other historical intervals were recapitulated or reversed during PACE trajectories, indicating that these states are compatible with BCL-2 family protein functions, but these substitutions could not have contributed to historical changes in PPI specificity, which remained unchanged on these branches. Our experiments therefore suggest strong effects of chance and contingency in the historical evolution of BCL-2's derived PPI specificity.

## Historical contingency is the major cause of sequence variation under selection for new functions

We next sought to directly quantify the relative effects of chance and historically generated contingency on sequence outcomes in our experiments. We analyzed the genetic variance – defined as the probability that two variable sites, chosen at random, are different in state – within and between trajectories from the same and different starting genotypes. To estimate the effects of chance, we compared the genetic variance between replicates initiated from the same starting genotype ($V_g$) to the within-replicate genetic variance ($V_r$). We found that $V_g$ was on average 30% greater than $V_r$, indicating that chance causes evolution to produce divergent genetic outcomes between independent lineages even with strong selection for a change in function (*Figure 5A*). We quantified contingency by comparing the pooled genetic variance among replicates from different starting genotypes ($V_t$) to that among replicates from the same starting genotype ($V_g$). Contingency's effect was even larger than that of chance, increasing $V_t$ by an average of 80% across all pairs of starting points compared to $V_g$ when selecting for a new function. Together, chance and contingency had a multiplicative effect, increasing the genetic variance among trajectories from different starting genotypes ($V_t$) by an average of 2.4-fold compared to the genetic variance within trajectories ($V_r$). The effects of

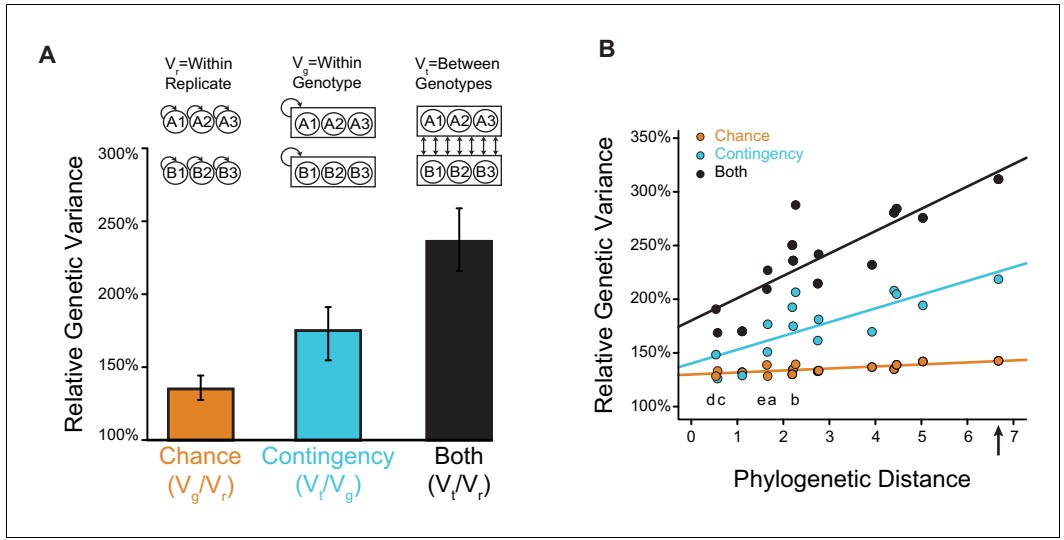

**Figure 5.** Effects of chance and contingency. (**A**) Variation in evolutionary sequence outcomes caused by chance (orange), contingency (teal), and both (black). Inset: schematic for estimating the effects of chance and contingency. Chance was estimated as the average genetic variance among replicates from the same starting genotype ($V_g$) divided by the within-replicate genetic variance ($V_r$). Contingency was estimated as the average genetic variance among replicates from different starting genotypes ($V_t$) divided by the average genetic variance among replicates from the same starting genotype ($V_g$). Combined effects of chance and contingency were estimated as the average genetic variance among replicates from different starting genotypes ($V_t$) compared to the within-replicate genetic variance ($V_r$). Genetic variance is the probability that two randomly drawn alleles are different in state. Error bars, 95% confidence intervals on the mean by bootstrapping PACE replicates. (**B**) Change in the effects of chance and contingency over phylogenetic distance. Each point is for a pair of starting proteins used for PACE, comparing the phylogenetic distance (the total length of branches separating them, in substitutions per site) to the effects of chance (orange), contingency (teal), or both (black), when PACE outcomes are compared between them. Solid lines, best-fit linear regression. Letters indicate the phylogenetic branch indexed in *Figure 3E*. The combined effect of chance and contingency increased significantly with phylogenetic distance (slope = 0.19, p=$2\times10^{-5}$), as did the effect of contingency alone (slope = 0.11, p=0.007). The effect of chance alone did not depend on phylogenetic distance (slope = 0.02, p=0.5). The combined effect of chance and contingency increased significantly faster than the effect of contingency alone (0.08, p=0.04). Arrow, phylogenetic distance between extant hsMCL-1 and hsBCL-2 proteins, which share AncMB1 as their most recent common ancestor.

The online version of this article includes the following figure supplement(s) for figure 5:

**Figure supplement 1.** Change in chance and contingency over time.

chance and contingency were not significantly different between PACE experiments in which protein interactions were gained and those in which they were lost (*Figure 5—figure supplement 1*).

The preceding analyses do not account for phylogenetic structure or the extent of divergence between starting points. We therefore assessed how chance and contingency changed with phylogenetic distance using linear regression (*Figure 5B*, *Figure 5—figure supplement 1*). We found that the effect of contingency on genetic variance increased significantly with phylogenetic divergence among starting points. The effect of chance did not increase with divergence, but the combined effect of contingency and chance increased even more rapidly than contingency alone because the total impact on genetic variance of these two factors is multiplicative by definition.

We next compared the impact of contingency to that of chance as phylogenetic divergence increases. On the timescale of metazoan evolution, contingency's effect (an increase in genetic variance by about 100%) was three times greater than that of chance when evolution was launched from extant starting points whose LCA was AncMB1, near the base of Metazoa (*Figure 5B*). The combined effect of chance and contingency on this timescale was a 3.2-fold increase in variance among single trajectories launched from these starting points. Even across the shortest phylogenetic intervals we studied, contingency's effect was larger than that of chance, although to a smaller extent. Taken together, these data indicate that contingency, magnified by chance, steadily increases the unpredictability of evolutionary outcomes as protein sequences diverge across history.

## Contingency is caused by epistasis between historical substitutions and specificity-changing mutations

Contingency is expected to arise in our experiments if historical substitutions (which separate ancestral starting points) interact epistatically with mutations that occur during PACE, causing the mutations that can confer selected PPI specificities to differ among starting points. To experimentally test this hypothesis and characterize underlying epistatic interactions, we first identified sets of candidate causal mutations that arose repeatedly during PACE replicates from each starting genotype. We then verified their causal effect on specificity by introducing only these mutations into the protein that served as the starting point for the PACE experiment in which they were observed and

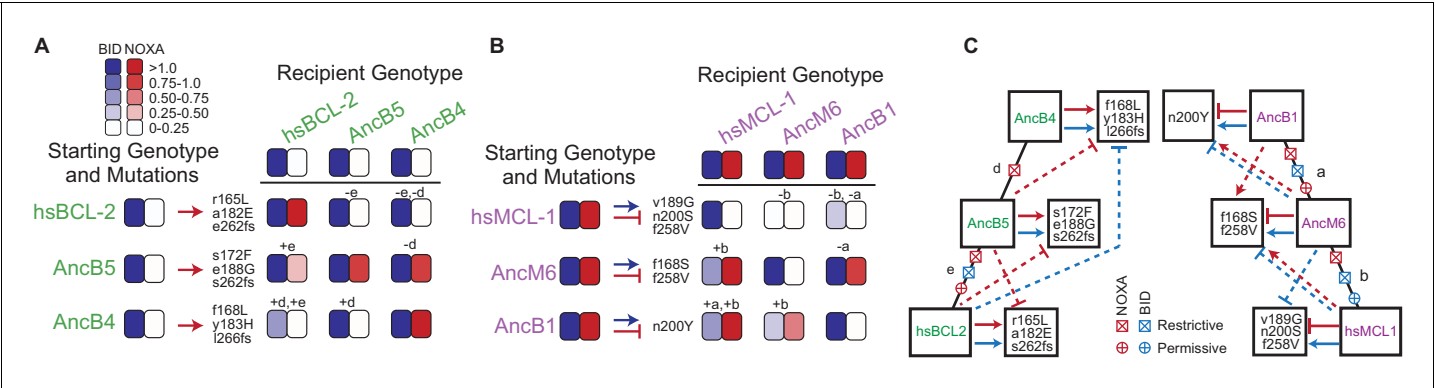

**Figure 6.** Sources of contingency. (A) Epistatic incompatibility of PACE mutations in other historical proteins. Effects on activity are shown when amino acid states acquired in PACE under selection to acquire NOXA binding (red arrows) are introduced into ancestral and extant proteins. The listed mutations that occurred during PACE launched from each starting point (rows) were introduced as a group into the protein listed for each column. Observed BID (blue) and NOXA (red) activity in the luciferase assay for each mutant protein are shown as heatmaps (normalized mean of three biological replicates). Letters indicate the phylogenetic branch in *Figure 3E* that connects the PACE starting genotype to the recipient genotype. Plus and minus signs indicate whether mutations were introduced into a descendant or more ancestral sequence, respectively. (B) Effects on activity when amino acids acquired in PACE under selection to lose NOXA binding and acquire BID binding are introduced into different ancestral and extant proteins, represented as in (A). (C) Epistatic interactions between historical substitutions and PACE mutations. Restrictive historical substitutions (X) cause mutations that alter PPI specificity in an ancestor to abolish either BID (blue) or NOXA (red) activity when introduced into later historical proteins. Permissive substitutions (+) cause PACE mutations that alter PPI specificity in a descendent to abolish either BID or NOXA activity in an ancestor. Arrow, gain or maintenance of binding. Blunt bar, loss of binding. Mutations that confer selected functions in PACE are shown in the boxes at the end of solid arrows or bars. Solid lines, functional changes under PACE selection. Dashed lines, functional effects different from those selected for when PACE-derived mutations are placed on a different genetic background.

measuring their effects on BID and NOXA binding. We found that all sets were sufficient to confer the selected-for specificity in their 'native' background (*Figure 6A, B*).

We then introduced these mutations into the other starting proteins that had been subject to the same selection regime and performed the same assay (*Figure 6A,B*). Eleven of 12 such swaps failed to confer the PPI specificity on other proteins that they conferred in their native backgrounds. These swaps compromised binding of BID, failed to confer the selected-for gain or loss of NOXA binding, or both. The only case in which the mutations that conferred the target phenotype during directed evolution had the same effect in another background was the swap into AncB4 of mutations that evolved in AncB5 – the most similar genotypes of all pairs of starting points in the analysis. Contingency therefore arose because historical substitutions that occurred during the intervals between ancestral proteins made specificity-changing mutations either deleterious or functionally inconsequential when introduced into genetic backgrounds that existed before or after those in which the mutations occurred.

To characterize the timing and effect of these epistatic substitutions during historical evolution, we mapped the observed incompatibilities onto the phylogeny (*Figure 6C*). We inferred that restrictive substitutions evolved on a branch if mutations that arose during directed evolution of an ancestral protein compromised coregulator binding when swapped into descendants of that branch. Conversely, we inferred that permissive substitutions evolved on a branch if mutations that arose during directed evolution compromised coregulator binding when swapped into more ancient ancestral proteins.

We found that both permissive and restrictive epistatic substitutions occurred on almost every branch of the phylogeny and affected both BID and NOXA binding. The only exception was the branch from AncB4 to AncB5, on which only restrictive substitutions affecting NOXA binding occurred. This is the branch immediately after NOXA function changed during history; it is also the shortest of all branches examined and the one with the smallest effect of contingency on genetic variance (*Figure 5B*). Even across this branch, however, the PACE mutations that restore the ancestral PPI specificity in AncB4 can no longer do so in AncB5. These results indicate that the paths through sequence space leading to historical PPI specificities changed repeatedly during the BCL-2 family's history, even during intervals when the proteins' PPI binding profiles did not evolve.

## Chance is caused by degeneracy in sequence–function relationships

For chance to strongly influence the outcomes of adaptive evolution, multiple paths to a selected phenotype must be accessible with similar probabilities of being taken. This situation could arise if several different mutations (or sets of mutations) can confer a new function or if mutations that have no effect on function accompany function-changing mutations by chance. To distinguish between these possibilities, we measured the functional effect of different sets of mutations that arose in replicates when hsMCL-1 was evolved to lose NOXA binding (*Figure 7A*, *Figure 7—figure supplement 1*). One mutation (v189G) was found at high frequency in all four replicates, but it was always accompanied by other mutations, which varied among trajectories. We found that v189G was a major contributor to the loss of NOXA binding, but it had this effect only in the presence of the other mutations, which did not decrease NOXA binding on their own. Mutation v189G therefore required permissive mutations to occur during directed evolution, and there were multiple sets of mutations with the potential to exert that effect; precisely which permissive mutations occurred in any replicate was a matter of chance. All permissive mutations were located near the NOXA binding cleft, suggesting a common mechanistic basis (*Figure 7B*).

Other starting genotypes showed a similar pattern of multiple sets of mutations capable of conferring the selected function (*Figure 7—figure supplement 2*). In addition, when mapped onto the protein structure, all sites that were mutated in more than one replicate either directly contacted the bound peptide or were on secondary structural elements that did so (*Figure 7C–D*), suggesting a limited number of structural mechanisms by which PPIs can be altered. Taken together, these results indicate that chance arose because from each starting genotype, there were multiple mutational paths to the selected specificity; partial determinism arose because the number of accessible routes was limited by the structure-function relationships required for peptide binding in this family of proteins.

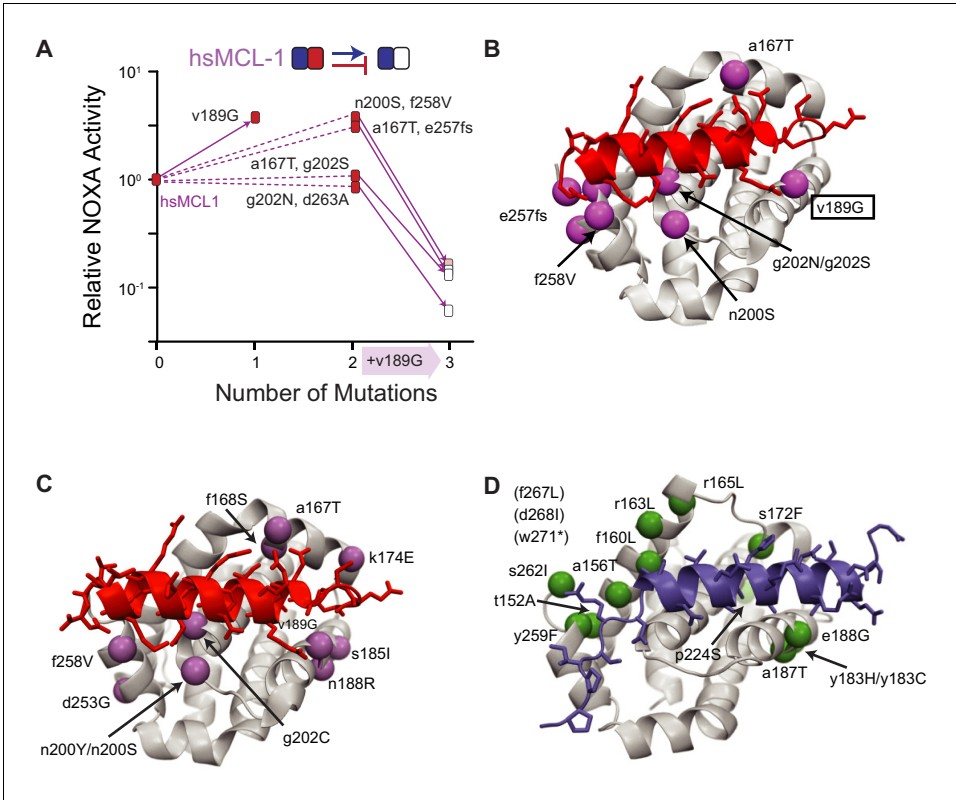

**Figure 7.** Sources of chance. (**A**) Dissecting the effects of sets of mutations (white boxes) that caused hsMCL-1 to lose NOXA binding during four PACE trajectories. Filled boxes show the effect of introducing a subset of mutations into hsMCL-1 (normalized mean relative from three biological replicates). Solid lines show the effect of introducing v189G, which was found in all four sets. Dotted lines, effects of the other mutations in each set. (**B**) Structural location of mutations in (**A**). Alpha-carbon atom of mutated residues are shown as purple spheres on the structure of MCL-1 (light gray) bound to NOXA (red, PDB 2nla). (**C**) Location of repeated mutations when hsMCL-1, AncM6, and AncB1 were selected to lose NOXA binding (purple spheres), represented on the structure of MCL-1 (gray) bound to NOXA (red, PDB 2nla). (**D**) Location of repeated mutations when hsBCL-2, AncB5, and AncB4 were selected to gain NOXA binding (green spheres), on the structure of hsBCL-xL (gray) bound to BID (blue, PDB 4qve).

The online version of this article includes the following figure supplement(s) for figure 7:

**Figure supplement 1.** Effects on NOXA binding of hsMCL-1 PACE-derived mutations.

**Figure supplement 2.** Phenotypic effects of reverting frequent PACE-derived mutations.

## Partial determinism is attributable to a limited number of function-changing mutations

We next analyzed the genetic basis for the limited degree of determinism that we observed in our experiments. Specifically, we sought to distinguish whether, from a given BID-specific starting point, only a few genotypes can confer NOXA binding while retaining BID binding or, alternatively, whether there are many such genotypes, but under strong selection a few are favored over others.

We performed PACE experiments in which we selected hsBCL-2 to retain its BID binding, without selection for or against NOXA binding; we then screened for variants that fortuitously gained NOXA binding using an activity-dependent plaque assay (**Figure 8A–B**). All four replicate populations produced clones that neutrally gained NOXA binding at a frequency of ~0.1% to 1% – lower than when NOXA binding was directly selected for but five orders of magnitude higher than when NOXA binding was selected against (**Figure 8A**, **Figure 8—figure supplement 1**). From each replicate, we then sequenced three NOXA-binding clones and found that all but one of them contained mutation r165L (**Figure 8B**), which also occurred at high frequency when the same protein was selected to gain NOXA binding (**Figure 8—figure supplement 2**). We introduced r165L into hsBCL-2 and found

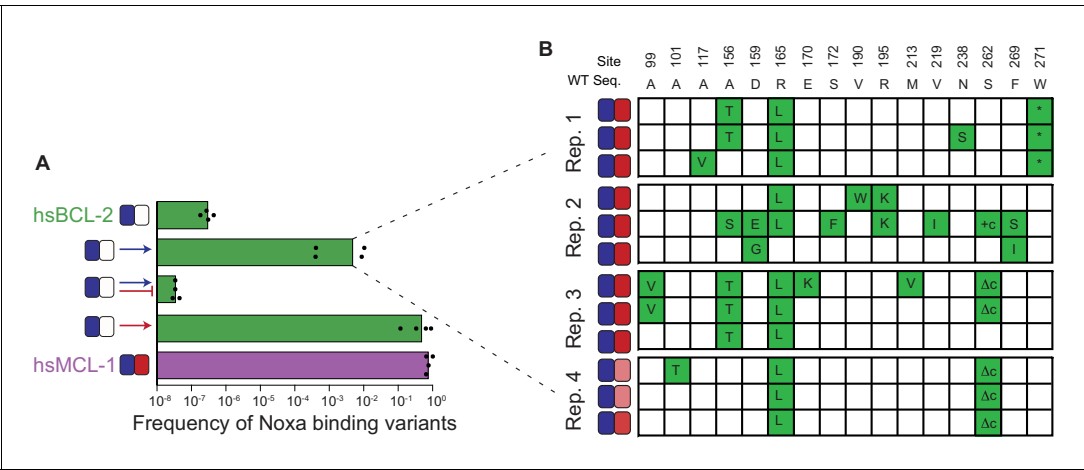

**Figure 8.** Sources of determinism. (**A**) Evolution of NOXA-binding phage under various selection regimes. Frequency was calculated as the ratio of plaque forming units (PFU) per milliliter on *E. coli* cells that require NOXA binding to the PFU on cells that require BID binding to form plaques. Wild-type hsBCL-2 (green) and hsMCL-1 (purple) are shown as controls. Arrow, positive selection for function. Bar, counterselection against function. Blue, BID. Red, NOXA. Bars are the mean of four trajectories for each condition (points). (**B**) Phenotypes and genotypes of hsBCL-2 variants that evolved NOXA binding under selection to maintain only BID binding. Sites and WT amino state are indicated at top. For each variant, non-WT states acquired are shown in green. Heatmaps show binding to BID and NOXA in the luciferase assay for each variant (normalized mean of three biological replicates). The online version of this article includes the following figure supplement(s) for figure 8:

**Figure supplement 1.** Selection schemes and phage titers for fortuitous NOXA binding of hsBCL2.
**Figure supplement 2.** Allele frequency of non-wt states during PACE.
**Figure supplement 3.** Effect on NOXA binding of the key r165L mutation.
**Figure supplement 4.** Selection and phage titers for fortuitous NOXA binding of AncB4 and AncB5.

that it conferred significant NOXA binding with little effect on BID binding (*Figure 8—figure supplement 3*). Several other mutations appeared repeatedly in clones that fortuitously acquired NOXA binding, and these mutations were also acquired under selection for NOXA binding (*Figure 8B*, *Figure 8—figure supplement 2*). A similar pattern of common mutations was observed in AncB4 and AncB5 clones that fortuitously or selectively evolved NOXA binding (*Figure 8—figure supplement 4*). These observations indicate that the partial determinism we observed arises because from these starting points only a few mutations have the potential to confer NOXA binding while retaining BID binding.

## Contingency can affect accessibility of new functions

Although we found that chance and contingency strongly influenced sequence outcomes in our experiments, all trajectories acquired the historically relevant PPI specificities that were selected for, indicating strong necessity at the level of protein function. This was true whether evolution began from more 'promiscuous' starting points

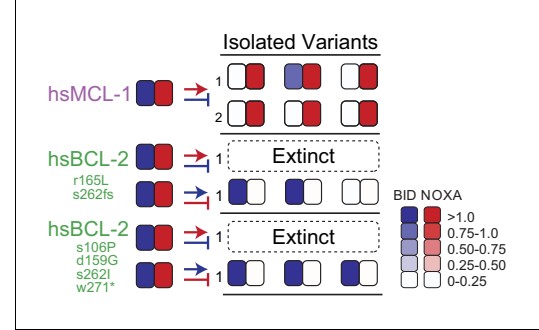

**Figure 9.** Contingency affects the evolution of novel specificity. Starting genotypes that can bind both BID and NOXA (left) were selected to lose only BID or NOXA binding. Heatmaps show binding to BID and NOXA in the luciferase assay for each starting genotype (on the left) and for three individual variants picked at the end of one or more PACE trajectories (index numbers). Each box displays the normalized mean of three biological replicates for one variant. Trajectories initiated from starting points produced by PACE (green) and then selected for a non-historical function (loss of BID binding) went extinct .
The online version of this article includes the following figure supplement(s) for figure 9:

**Figure supplement 1.** Selection scheme and phage titers for the gain of NOXA specificity.
**Figure supplement 2.** Selection scheme and phage titers for the regain of BID specificity.

that bound both BID and NOXA or from more specific proteins that bound only BID.

To further probe the evolutionary accessibility of new functions, we used PACE to select for a PPI specificity that never arose during historical evolution – binding of NOXA but not BID. We found that trajectories launched from hsMCL-1 (which binds both coregulators) readily evolved the selected phenotype, but two PACE-evolved variants of hsBCL-2, which had acquired the same PPI profile as hsMCL-1, went extinct under the same selection conditions (*Figure 9*, *Figure 9—figure supplement 1*). The inability of the derived hsBCL-2 genotypes to acquire NOXA specificity was not attributable to a general lack of functional evolvability by these proteins because they successfully evolved in a separate PACE experiment to lose their NOXA binding but retain BID binding (*Figure 9—figure supplement 2*). These results establish that contingency can influence the accessibility of new functions and that the sequence by which a specific functional phenotype is encoded can play important roles in subsequent phenotypic evolution.

## Discussion

The two major paradigms of 20th-century evolutionary biology – the adaptationist program (*Mayr, 1983*) and the neutral theory of molecular evolution (*Kimura, 1986*) – focus on either necessity or chance, respectively, as the primary mode of causation that produces natural variation in molecular sequences. Neither of these schools of thought admits much influence from contingency or history. From an adaptationist perspective, variation is caused by natural selection, which generates optimal forms under different environmental conditions. Differences in protein sequence or other properties are interpreted as the result of adaptive changes that improved a molecule's ability to perform its function in the species' particular environment (*Goodsell and Olson, 2000*; *Nguyen et al., 2017*; *Somero, 1995*; *Závodszky et al., 1998*). For neutralists, variation reflects the influence of chance in choosing among biologically equivalent possibilities, and conservation reflects purifying selection, both of which are viewed as largely unchanging across sequences in an alignment. For example, conserved portions of molecular sequences are interpreted as essential to structure and function, whereas differences in sequence alignments reflect a lack of constraint (*Echave et al., 2016*; *Kimura and Ohta, 1974*; *Perutz et al., 1965*). In neither worldview, does the particular state of a system strongly reflect its past or shape its evolutionary future. Recent work has shown that contingency might athe sequence outcomes of evolution (*Bloom et al., 2010*; *Blount et al., 2012*; *Blount et al., 2008*; *Breen et al., 2012*; *Bridgham et al., 2009*; *Ortlund et al., 2007*; *Pollock et al., 2012*; *Quandt et al., 2015*; *Sailer et al., 2017*; *Shah et al., 2015*; *Starr et al., 2018*), echoing themes raised in paleontology (*Gould, 1989*; *Jablonski, 2017*) and developmental biology (*Gompel et al., 2005*; *Shubin et al., 2009*). Despite these recent findings, the dominance of the adaptationist and neutralist worldviews – and the continuing rhetorical battle between them (*Jensen et al., 2019*; *Kern and Hahn, 2018*) – has obscured the possibility that contingency might join selection, drift, and mutation as a primary factor shaping the outcomes of evolution.

We found that contingency generated by sequence change over phylogenetic timescales plays a profound role in BCL-2 family protein sequence evolution under laboratory selection for new functions. The mutations that rose to high frequency during experimental evolution were almost completely different among evolutionary trajectories initiated from historical starting points separated by long phylogenetic distances. We observed a strong role for chance (because trajectories launched from the same starting point evolved extensive differences from each other) and an even greater effect of contingency (because pools of trajectories launched from different starting genotypes evolved even greater differences). When combined, chance and contingency erased virtually all traces of necessity between individual trajectories initiated from distantly related starting points. With the exception of a single truncation mutation that does not affect the selected-for function, the only predictable sequence states were those that remained unchanged from the starting point in all trajectories, presumably because they are unconditionally necessary for both PPIs tested and were therefore conserved by purifying selection.

Contingency and chance are distinct but interacting modes of causality; our experiments allowed us to disentangle their individual effects and interactions. By calculating genetic variance among replicates from the same starting point and among pooled replicates from different starting points, we quantified the effect of chance and contingency, respectively. The total effect of chance and contingency together – genetic variance among replicates from different starting points – is by definition

the product of the separate effects of chance and contingency. This quantitative relationship reflects the intrinsic interaction between chance and contingency in evolutionary processes (*Beatty and Carrera, 2011*; *Desjardins, 2011*). At any point in history, numerous sets of mutations were accessible, and chance determined which ones occurred. These chance events then determined the steps that could be taken during future intervals, because of contingency. Without chance, contingency – dependence of the accessibility of future trajectories on the protein's state – would never be realized or observed: all phylogenetic lineages launched from a common ancestor would always lead to the same intermediate steps and thus the same ultimate outcomes. Conversely, without contingency, chance events would have no impact on the accessibility of other mutations because every path that was ever open would remain forever so, irrespective of the random events that happen to take place. The outcomes of evolution from a common ancestral starting point are therefore unpredictable when intermediate steps shape future possibilities (contingency), and those intermediate steps cannot be predicted because multiple possibilities are accessible at any point in history (chance).

Our experimental design approximates but does not quite achieve the ideal design of multireplicate evolution from ancestral starting points under historical conditions, because the conditions we imposed during PACE differ in several ways from those that pertained during historical evolution. Many factors that give rise to chance, contingency, and necessity are likely to be similar between history and our experiments. For example, factors related to a protein's sequence–structure–function relations – such as the number of mutations that can produce a particular function and the nature of epistasis among them – play a key role in chance and contingency and are shared between PACE and history. Other aspects of our design may underestimate the effects of chance and contingency during history. For example, the population genetic parameters in our experimental conditions favor determinism because they involve very large population sizes, strong selection pressures, and high mutation rates, all directed at a single gene. If population sizes during historical BCL-2 family evolution involved smaller populations, weaker selection, lower mutation rates, and a larger genetic 'target size' for adaptation, as seems likely, then chance would have played an even larger role during history than in our experiments. In addition, we used human BID and NOXA as fixed binding partners, but during real evolution these proteins would have varied in sequence as well, introducing opportunities for chance and contingency to further affect the sequence outcomes of BCL-2 evolution.

Some differences between our design and the biological setting of historical BCL-2 family evolution could have overestimated chance's historical role. We selected for PPI interactions with two particular peptides, leaving out many potential cellular binding partners. PACE takes place in the cytosol of *E. coli* cells, but BCL-2 evolution occurred in animal cells, and natural BCL-2 proteins are partially membrane-bound. These additional dimensions of BCL-2 biological function could have imposed additional selective constraints on the evolution of BCL-2 family proteins historically, reducing the number of functionally equivalent genotypes available to chance. We used peptide fragments from coregulator proteins rather than full-length BID and NOXA; however, the peptide-binding cleft is cytosolic, and recent work indicates that relative affinity of BCL-2 family proteins is similar between peptides and full-length coactivators, although absolute affinity is typically higher in the latter case (*Kale et al., 2018*). Whether these differences quantitatively affect chance and contingency in PACE versus historical evolution is unknown. Finally, because our experimental design imposed selection for new PPI specificities, it does not reveal the effects of chance and contingency under different selective regimes, such as purifying selection to maintain an existing function, which may or may not be similar.

We studied a particular protein family as a model, but we expect that qualitatively similar results may apply to many other proteins. Epistasis is a common feature of protein structure and function, so the accumulating effect of contingency across phylogenetic time in the BCL-2 family will probably be a general feature of protein evolution, although its rate and extent are likely to vary among protein families and timescales (*Chandler et al., 2013*; *Harms and Thornton, 2014*; *Shah et al., 2015*; *Zhu et al., 2018*). The influence of chance depends upon the existence of multiple mutational sets that can confer a new function; this kind of degeneracy is likely to pertain in many cases: greater determinism is expected for functions with very narrow sequence–structure–function constraints, such as catalysis (*Hawkins et al., 2018*; *Karageorgi et al., 2019*; *Menéndez-Arias, 2010*; *Meyer et al., 2012*; *Salverda et al., 2011*; *Storz, 2016*), than those for which sequence requirements are less strict, such as substrate binding (*Blount et al., 2012*; *Starr et al., 2017*;

*Yokoyama et al., 2008*; *Zheng et al., 2019*). Consistent with this prediction, when experimental evolution regimes have imposed diffuse selection pressures on whole organisms, making loci across the entire genome potential sources of adaptive mutations, virtually no repeatability has been observed among replicates (*Kryazhimskiy et al., 2014*; *Wünsche et al., 2017*).

The method that we developed for rapid evolution of PPI specificity has several advantages that can be extended to other protein families. First, by using PACE, many replicates can be evolved in parallel across scores or hundreds of generations in just days, with minimal need for intervention by the experimentalist (*Esvelt et al., 2011*). Second, our split RNAP design for acquiring new PPIs has fewer components than previous methods for this purpose, such as two-hybrid designs; this makes it considerably easier to tune and optimize and therefore to extend to other protein systems. Third, unlike approaches that attempt to evolve specific PPIs by alternating selection and counterselection through time, our platform simultaneously imposes selection and counterselection within the same cell, thus selecting for specificity directly. By combining these elements in a single system, our platform should allow rapid multireplicate evolution of new cytosolic PPI specificities in a variety of protein families.

Our results have implications for efforts to engineer proteins with desired properties. We found no evidence that ancestral proteins were more or less 'evolvable' than extant proteins: the selected-for phenotypes readily evolved from both extant and ancestral proteins with the same starting binding capabilities. Moreover, chance's effect was virtually constant across ~1 billion years of evolution, indicating that the number of accessible mutations in the deep past that could confer a selected-for function was apparently no greater than it is now. Nevertheless, the strong effect of contingency that we observed on sequence evolution – and its partial role in the acquisition of new functions per se – suggests that efforts to produce proteins with new functions by design or directed evolution will be most effective and will lead to more diverse sets of sequence outcomes, if they use multiple different protein sequences as starting points, ideally separated by long intervals of sequence evolution. Ancestral proteins can be useful for this purpose simply because they provide routes to functions that were inaccessible from extant protein, even if those routes are not fundamentally different in number or kind.

Finally, our work has implications for understanding the processes of protein evolution and the significance of natural sequence variation. Our observations suggest that sequence–structure–function associations apparent in sequence alignments are to a significant degree the result of contingent constraints that were transiently imposed or removed by chance events during history (*Gong et al., 2013*; *Harms and Thornton, 2014*; *Starr et al., 2018*; *Starr et al., 2017*). Evolutionary explanations of sequence diversity and conservation must therefore explicitly consider the historical trajectories by which sequences evolved, in contrast to the largely history-free approaches of the dominant schools of thought in molecular evolution. Our findings suggest that present-day BCL-2 family proteins – and potentially many others, as well – are largely physical anecdotes of their particular unpredictable histories: their sequences reflect the interaction of accumulated chance events during descent from common ancestors with necessity imposed by physics, chemistry, and natural selection. Apparent 'design principles' in the pattern of variability and conservation in extant proteins reflect not how things must be to perform their functions, or even how they can best do so. Rather, today's proteins reflect the legacy of opportunities and limitations that they just happen to have inherited.

## Materials and methods

**Key resources table**

| Reagent type (species) or resource | Designation | Source or reference | Identifiers | Additional information |
|---|---|---|---|---|
| Strain, strain background (*Escherichia coli*) | S1030 | *Carlson et al., 2014* | | |
| Strain, strain background (*Escherichia coli*) | 1059 | *Carlson et al., 2014* | | |

*Continued on next page*

*Continued*

| Reagent type (species) or resource | Designation | Source or reference | Identifiers | Additional information |
|---|---|---|---|---|
| Strain, strain background (*Escherichia coli*) | NEB 10-beta | NEB | Cat# C3019I | |
| Strain, strain background (*Escherichia coli*) | BCL21 (DE3) | NEB | Cat# C2530H | |
| Peptide, recombinant protein | BID | GenScript | This Study | Human BID peptide used for fluorescence polarization (see Materials and methods) |
| Peptide, recombinant protein | NOXA | Genscript | This study | Human NOXA peptide used for fluorescence polarization (see Materials and methods) |
| Commercial assay or kit | DNA clean and concentrator kit | Zymo | Cat# D4013 | |
| Commercial assay or kit | MiSeq Reagent Kit v3 | Illumina | Cat# MS-102–3003 | |
| Chemical compound, drug | Q5 DNA Polymerase | NEB | Cat# M0491 | |
| Chemical compound, drug | Phusion DNA polymerase | ThermoFisher Scientific | Cat# F518L | |
| Chemical compound, drug | Isopropyl-b-D-thiogalactopyranoside (IPTG) | bioWORLD | Cat# 21530057 | |
| Chemical compound, drug | His60 Ni Superflow Resin | Takara | Cat# 635660 | |
| Software, algorithm | Geneious | Geneious | 10.1.3 | |
| Software, algorithm | R | CRAN | 3.5.1 | |
| Software, algorithm | RStudio | RStudio | 1.1.456 | |
| Software, algorithm | PROT Test | *Abascal et al., 2005* | 3.4.2 | |
| Software, algorithm | RAXML-ng | *Kozlov and Stamatakis, 2019* | 0.6.0 | |

## Phylogenetics

Amino acid sequences of the human BCL-2, BCLW, BCL-xL, MCL-1, NRH, BFL1, BAK, BAX, and BOK paralogs were used as starting points for identifying BCL-2 family members in other species. For each paralog, tblastn and protein BLAST on NCBI BLAST were used to identify orthologous sequences between January and March of 2018 (*Altschul et al., 1997*). Sequences for each paralog were aligned using MAFFT (G-INS-I) with the –allowshift option and –unalignlevel set at 0.1. For each paralog, phylogenetic structure was determined using fasttree 2.1.11 within Geneious 10.1.3. Missing clades based on known species relationships were then identified, and specific tblastn searches were used within Afrotheria (taxid:311790), Marsupials (taxid:9263), Monotremes (taxid:9255), Squamata (taxid:8509), Archosauria (taxid:8492), Testudinata (taxid:8459), Amphibia (taxid:8292), Chondrichthyes (taxid:7777), Actinopterygii (taxid:7898), Dipnomorpha (taxid:7878), Actinistia (taxid:118072), Agnatha (taxid:1476529), Cephalochordata (taxid:7735), and Tunicata (taxid:7712) as needed. Additional sequences were added by downloading genome and transcriptome data for tuatara (*Miller et al., 2012*), sharks and rays (*Wyffels et al., 2014*), gar (*Zerbino et al., 2018*), ray-finned fish (*Hughes et al., 2018*), lamprey (*Smith et al., 2018*), hagfish (*Takechi et al., 2011*), *Ciona savignyi* (*Zerbino et al., 2018*), tunicates (*Delsuc et al., 2018*), echinoderms (*Reich et al., 2015*), porifera (*Riesgo et al., 2014*), and ctenophores (*Moroz et al., 2014*). In each case, local BLAST databases were created in Geneious and searched using tblastn. Finally, we used BCL-2DB to add missing groups as needed (*Rech de Laval et al., 2014*).

After collection of sequences, each paralog was realigned using MAFFT (G-INS-I) with the –allowshift option and –unalignlevel set at 0.1. Based on known species relationships, lineage-specific insertions were removed and gaps manually edited. Only a single sequence was kept among pairs of sequences differing by a single amino acid and sequences with more than 25% of missing sites were removed. For difficult to align sequences, sequences were modeled on the structures of human BCL-

2 family members using SWISS-Model to identify likely locations of gaps (*Waterhouse et al., 2018*). Finally, paralogs were profile aligned to each other, and paralog-specific insertions were identified.

In total, 151 amino acid sites from 745 taxa were used to infer the phylogenetic relationships among BCL-2 family paralogs. PROT Test 3.4.2 was used to identify the best-fit model among JTT, LG, and WAG, with combinations of observed amino acid frequencies (+F), gamma distributed rate categories (+G), and an invariant category (+I) (*Abascal et al., 2005*). From this, JTT + G + F had the highest likelihood and lowest Aikake Information Criterion score. RAXML-ng 0.6.0 was then used to identify the maximum likelihood tree using JTT+G12+F0 (12 gamma rate categories with maximum likelihood estimated amino acid frequencies) (*Kozlov and Stamatakis, 2019*). Finally, we enforced monophyly within each paralog for the following groups: lobe-finned fish (n = 9), ray-finned fish (n = 9), jawless fish (n = 5), cartilaginous fish (n = 8), tunicates (n = 4), branchiostoma (n = 4), chordates (n = 5), ambulacraria (n = 5, hemichordata +echinodermata), deuterostomia (n = 5), protostomia (n = 5), cnidaria (n = 5), and porifera (n = 4) (values in parenthesis are number of identified paralogs in each group) and used RAXML-ng with JTT + G12 + F0 to identify the best tree given these constraints (Supplementary Data Phylogenetic.Data.zip).

Overall, we recovered three clades: a pro-apoptotic clade; a clade containing the BCL-2, BCLW, and BCLX vertebrate paralogs and BCL non-vertebrate sequences; and a clade containing the MCL-1, BFL1, and NRH vertebrate paralogs and MCL non-vertebrate sequences. We used the pro-apoptotic clade as the outgroup to the two anti-apoptotic clades. Within the BCL-2 clade, the majority of vertebrates contained all three copies. However, the exact relationship among the paralogs was unclear; only two copies were identified within jawless fish and their phylogenetic placement had weak support. Non-vertebrate clades tended to have good support and only a single copy. However, support for these groups following established species relationships was often limited. The MCL-1 clade contained the fastest evolving paralogs of the BCL-2 family. As with the BCL-2-like clade, only two copies were found within the jawless fish and the exact sister relationships among paralogs was unclear. Non-vertebrates contained only a single copy, but as with the BCL-2-like clade, support for relationships following established species relationships was often weak.

The BCL-2-like and MCL-1-like paralogs formed a clade with the BHP1 and BHP2 sequences from porifera. The sister relationships among these four clades were unresolved. In addition, we recovered a sister relationship between the BAK and BAX paralogs. While both paralogs contained copies from porifera, these clades evolved quickly and had relatively low support, and they may be artifactual. We identified only a single clade of ctenophores. Finally, the placement of BOK was unresolved; BOK may be sister to the BAK/BAX clade or an outgroup to all clades and the most ancient copy of the BCL-2 family.

## Ancestral reconstruction

Posterior probabilities of each amino acid at each site were inferred using Lazarus (*Finnigan et al., 2012*) to run codeml within PAML. We used the same model and alignment as used to infer the phylogeny. We used the branch lengths and topology of the constrained maximum likelihood phylogeny found by raxml-ng.

We first reconstructed the LCAs of all BCL-2 and MCL-1 like sequences, AncMB1-M, using the maximum likelihood state for each alignable site. We then reconstructed a series of ancestors from AncMB1 to modern human MCL-1. These included AncM1 (LCA of MCL-1-related sequences), AncM2 (LCA of MCL-1- related deuterostomes and protostomes), AncM3 (LCA of MCL-1-related deuterostomes), AncM4 (LCA of MCL-1-related urochordates and chordates), AncM5 (LCA of MCL-1, BFL1, and NRH like copies in vertebrates), AncM6 (LCA of MCL-1 and BFL1 like copies), AncMCL-1 (LCA of MCL-1 like copies), AncMCL-1-G (LCA of MCL-1 like Gnathostomes), AncMCL-1-O (LCA of MCL-1 like Osteichthyes), and AncMCL-1-T (LCA of MCL-1 like Tetrapods), AncMCL-1-A (LCA of MCL-1 like Amniotes), and AncMCL-1-M (LCA of MCL-1 like Mammals). In each case, the sequence of each ancestor used the maximum likelihood state at each site, with gaps inserted based on parsimony. We used the modern sequences of human MCL-1 to fill in portions of the sequence that showed poor alignment and could not be reconstructed, including both the N and C terms, as well as the loop between the first and second alpha helices. Average posterior probabilities for ancestors in the MCL-1 clade ranged from 0.73 (AncM6) to 0.98 (AncMCL-1-M) with an average of 0.83 (sd 0.08) (*Supplementary file 2*).

For the BCL-2 like clade, we also reconstructed AncMB1, this time using human BCL-2 sequence to fill in the N and C terms and the loop between the first and second alpha helices (AncMB1-B). We then reconstructed sequences from AncMB1 to modern human BCL-2. These included AncB1 (LCA of BCL-2-related sequences), AncB2 (LCA of BCL-2-related Bilaterian and Cnidaria), AncB3 (LCA of BCL-2-related deuterostomes and protostomes), AncB4 (LCA of BCL-2 deuterostomes), AncB5 (LCA of BCL-2, BCLW, and BCLX like copies in vertebrates), AncBCL-2 (LCA of BCL-2 like copies), AncBCL-2-G (LCA of BCL-2 like gnathostomes), AncBCL-2-O (LCA of BCL-2 like osteichthyes), and AncBCL-2-T (LCA of BCL-2 like tetrapods), using human BCL-2 sequences for the N and C terms and the loop between the first and second alpha helices. Average posterior probabilities for ancestors in the BCL-2 clade ranged from 0.87 (AncB1) to 0.95 (AncBCL-2-T) with an average of 0.9 (sd 0.04).

## Test of robustness of ancestral inference

To determine the robustness of our conclusions on the phenotype of ancestral sequences, we synthesized and cloned alternative reconstructions for key ancestors. In each case, sequences contained the most likely alternative state with posterior probability > 0.2 for all such sites where such a state existed. Alternative reconstructions contained an average of 24 alternative states and represent a conservative test of function (min: 4, max: 44, *Supplementary file 2*). In our luciferase assay, all but two alternative reconstructions retained similar BID and NOXA binding as the maximum likelihood ancestral sequences. The first alternative reconstruction that differed from the maximum likelihood reconstruction was AltAncB3, which bound both BID and NOXA, while the ML for AncB3 bound BID, but NOXA only weakly. As a result, the exact branch upon which NOXA binding was lost historically is not resolved by this data.

The second alternative reconstruction that differed from the ML reconstruction was AltAncMB1-B, which had weaker NOXA binding than the ML reconstruction. To further test the robustness of AncMB1-B to alternative reconstructions, we synthesized and tested additional reconstructions that included only alternative amino acids with posterior probabilities greater than 0.4 (n = 3), 0.35 (n = 7), 0.3 (n = 13), and 0.25 (n = 18), and compared these to AncMB1-B and the 0.2 AltAncMB1-B (n = 21) (values in parentheses are number of states that differ from the ML state). We found that the 0.4, 0.35, and 0.3 alternative reconstructions bound both BID and NOXA, while the 0.25 and 0.2 alternative reconstructions had diminished NOXA binding.

Finally, we synthesized and tested modern sequences from key groups to determine the robustness of our inference on the timing of NOXA binding loss. These included BCL-2-related sequences from groups that diverged prior to the predicted loss of NOXA binding (*Trichoplax adhaerens* and *Hydra magnapapillata*), sequences from groups that diverged around the time of predicted NOXA binding loss (*Octopus bimaculoides* and *Stegodyphus mimosarum*), or sequences from groups predicted to have diverged after NOXA binding lost (*Saccoglossus kowalevskii* and *Branchiostoma belcheri*). In each case, we used human BCL-2 sequence to replace extant N and C terms and the loop between the first and second alpha helices. The *T. adhaerens* and *B. belcheri* sequences were nonfunctional in our luciferase assays, binding neither BID nor NOXA. However, recent work has comprehensively characterized binding in BCL-2 family members within *T. adhaerens*, finding that the BCL copy can bind both BID and NOXA as predicted (*Popgeorgiev et al., 2020*). *H. magnapapillata* bound both BID and NOXA in our assay and the remaining sequences bound only BID, suggesting a loss of NOXA binding prior to the divergence of protostomes and deuterostomes in the BCL-2 related clade, consistent with the conclusion drawn using reconstructed proteins.

## *Escherichia coli* strains

*E. coli* 10-beta cells were used for cloning and were cultured in 2xYT media. *E. coli* BL21 (BE3) cells were used for protein expression and were cultured in Luria-Bertain (LB) broth. *E. coli* S1030 cells cultured in LB broth were used for activity-dependent plaque assays, phage growth assays, and luciferase assays. S1030 cells cultured in Davis Rich media were used for PACE experiments (*Carlson et al., 2014*). *E. coli* 1059 cells were used for cloning phage and assessing phage titers and were cultured in 2xYT media.

## Cloning and general methods

Plasmids were constructed by using Q5 DNA Polymerase (NEB) to amplify fragments that were then ligated via Gibson Assembly. Primers were obtained from IDT, and all plasmids were sequenced at the University of Chicago Comprehensive Cancer Center DNA Sequencing and Genotyping Facility. Vectors and gene sequences used in this study are listed in *Supplementary file 5*, with links to fully annotated vector maps on Benchling. Key vectors are deposited at Addgene, and all vectors are available upon request. The following working concentrations of antibiotics were used: 50 µg/mL carbenicillin, 50 µg/mL spectinomycin, 40 µg/mL kanamycin, and 33 µg/mL chloramphenicol. Protein structures and alignments were generated using the program PyMOL (*Schrödinger, 2018*).

## Luciferase assay

Cloned expression vectors contained the following: (1) a previously evolved, isopropyl β-D-1-thiogalactopyranoside (IPTG)-inducible N-terminal half of T7 RNAP (*Zinkus-Boltz et al., 2019*) fused to a BCL-2 family protein; (2) the C-terminal half of T7 RNAP fused to a peptide from a BH3-only protein; and (3) T7 promoter-driven luciferase reporter. Chemically competent S1030 *E. coli* cells (*Carlson et al., 2014*) were prepared by culturing to an $OD_{600}$ of 0.3, washing twice with a calcium chloride/HEPES solution (60 mM $CaCl_2$, 10 mM HEPES pH 7.0, 15% glycerol), and then resuspending in the same solution. Vectors were transformed into chemically competent S1030 cells via heat shock at 42°C for 45 s, followed by 1 hr recovery in 3× volume of 2xYT media, and then plated on agar with the appropriate antibiotics (carbenicillin, spectinomycin, and chloramphenicol) to incubate overnight at 37°C. Individual colonies (three to four biological replicates per condition) were picked and cultured in 1 mL of LB media containing the appropriate antibiotics overnight at 37°C in a shaker. The next morning, 50 µL of each culture was diluted into 450 µL of fresh LB media containing the appropriate antibiotics, as well as 1 µM of IPTG. The cells were incubated in a shaker at 37°C, and $OD_{600}$ and luminescence measurements were recorded between 2.5 and 4.5 hr after the start of the incubation. Measurements were taken on a Synergy Neo2 Microplate Reader (BioTek) by transferring 150 µL of the daytime cultures into Corning black, clear-bottom 96-well plates. Data were analyzed in Microsoft Excel and plotted in GraphPad Prism, as previously reported (*Pu et al., 2017a*).

Protein expression hsBCL-2, hsMCL-1, and evolved variants were constructed as N-terminal 6xHis-GST tagged proteins. The recombinant proteins were expressed in BL21 *E. coli* (NEB) and purified following standard Ni-NTA resin purification protocols (ThermoFisher Scientific) (*Zhou et al., 2019*). Briefly, BL21 *E. coli* containing an N-terminal 6xHis-GST tagged BCL-2 family protein were cultured in 5 mL LB with carbenicillin overnight. The following day, the culture was added to 0.5 L of LB with carbenicillin, incubated at 37°C until it reached an $OD_{600}$ of 0.6, induced with IPTG (final concentration: 200 µM), and cultured overnight at 16°C. The cell pellet was harvested by centrifugation followed by resuspension in 30 mL of lysis buffer (50 mM Tris 1 M NaCl, 20% glycerol, 10 mM TCEP, pH 7.5) supplemented by protease inhibitors (200 nM Aprotinin, 10 µM Bestatin, 20 µM E-64, 100 µM Leupeptin, 1 mM AEBSF, 20 µM Pepstatin A). Cells were lysed via sonication and were then centrifuged at 12,000 g for 40 min at 4°C. Solubilized proteins, located in the supernatant, were incubated with His60 Ni Superflow Resin (Takara) for 1 hr at 4°C, and the protein was eluted using a gradient of imidazole in lysis buffer (50–250 mM). Fractions with the protein, as determined by SDS-PAGE, were concentrated in Ulta-50 Centrifugal Filter Units (Amicon, EMD Millipore). Proteins were purified via a desalting column with storage buffer (50 mM Tris–HCl [pH 7.5], 300 mM NaCl, 10% glycerol, 1 mM DTT) and further concentrated. The concentration of the purified BCL-2 family proteins was determined by BCA assay (ThermoFisher Scientific), and they were flash-frozen in liquid nitrogen and stored at −80°C.

## Fluorescent polarization binding assays

Fluorescent polarization (FP) was used to measure the affinity of BCL-2 family proteins with peptide fragments of the BH3-only proteins in accordance with previously described methods (*Zhang et al., 2002*). hsBCL-2, hsMCL-1, and evolved variants were purified as described above. The fluorescent NOXA and BID peptides (95+% purity) were synthesized by GenScript and were N-terminally labeled with 5-FAM-Ahx and C-terminally modified by amidation. These peptides were dissolved and stored in DMSO. Corning black, clear-bottom 384-well plates were used to measure FP, and three replicates were prepared for each data point. Each well contained the following 100 µL reaction: 20 nM

BH3-only protein, 0.05 nM to 3 µM of BCL-2 family protein (1/3 serial dilutions), 20 mM Tris (pH 7.5), 100 mM NaCl, 1 mM EDTA, and 0.05% pluronic F-68. FP values (in milli-polarization units; mFP) of each sample were read by a Synergy Neo2 Microplate Reader (BioTek) with the FP 108 filter (485/530) at room temperature 5–15 min after mixing all the components. Data were analyzed in Graph-Pad Prism 8, using the following customized fitting equation, to calculate $K_d$ (*Zhou et al., 2019*):

$$y = B + C\left(D + K_d + x - \sqrt{(D + K_d + x)^2 - 4Dx}\right)$$

where $y$ is normalized measured FP, $x$ is the concentration of BCL-2 protein, $D$ is the concentration of the BH3-only protein, $B$ and $C$ are parameters related to the FP value of free and bound BH3-only protein, and $K_d$ is the dissociation constant.

## Phage-assisted continuous evolution

PACE was used to evolve hsBCL-2, hsMCL-1, and ancestral proteins in accord with previously reported technical methods (*Carlson et al., 2014*; *Esvelt et al., 2011*; *Pu et al., 2019*; *Pu et al., 2017b*) using a new vector system. Briefly, combinations of accessory plasmids and the MP6 mutagenesis plasmid (*Badran and Liu, 2015*) were transformed into S1030 *E. coli*., plated on agar containing the appropriate antibiotics (carbenicillin, kanamycin, and chloramphenicol) and 10 mM glucose, and incubated overnight at 37°C. Colonies were grown overnight in 5 mL of LB containing the appropriate antibiotics and 20 mM glucose. Davis Rich media was prepared in 5–10 L carboys and autoclaved, and the PACE flasks and corresponding pump tubing were autoclaved as well. The following day, PACE was set up in a 37°C environmental chamber (Forma 3960 environmental chamber, ThermoFisher Scientific). For each replicate, an overnight culture was added to ~150 mL of Davis Rich carboy media in chemostats and grown for 2–3 hr until reaching an $OD_{600}$ of approximately 0.6. Lagoons containing 20 µL of phage from saturated phage stocks ($10^8$–$10^9$ phage) were then connected to the chemostat. Magnetic stir bars were used to agitate chemostats and lagoons. The chemostat cultures were flowed into the lagoons at a rate of approximately 20 mL/h. Waste output flow rates were adjusted to maintain a constant volume of 20 mL in the lagoons, 150 mL in the chemostat, and an $OD_{600}$ close to 0.6 in the chemostat. A 10% w/v arabinose solution was pumped into the lagoons at a rate of 1 mL/h. If the experiment included a mixing step (two separate chemostats flowed together into one lagoon for a mixed selection pressure), a chemostat was prepared the next day (as described above) and connected to the lagoons. During this step, lagoon volumes were increased to 40 mL, and the arabinose inflow rate was increased to 2 mL/h. After disconnecting the first chemostat the next day, the lagoon volumes and arabinose inflow were both lowered to 20 mL and 1 mL/h, respectively. During the experiment, samples were collected from the lagoons every 24 hr and centrifuged at 13,000 rpm for 3 min to collect the phage-containing supernatant, as well as the cell pellet for DNA extraction. PACE experiments are listed in *Supplementary file 3*. A single replicate of AncB5 was removed because of contamination. No statistical method was used to determine the number of replicates as only four independent replicate experiments could be performed simultaneously.

During PACE, the media volume of each lagoon turned over once per hour for 4 days, or ~100 times. For a phage population to survive this amount of dilution, a similar number of generations must have occurred between the starting phage and the phage in the lagoon at the end of the experiment (*Esvelt et al., 2011*). This is expected to be a conservative estimate; as a more fit phage rises in frequency in the population, it will undergo a greater number of generations than less-fit phage in the population. The mutagenesis plasmid MP6 induces a mutation rate of approximately $6 \times 10^{-6}$ per bp per generation. The BCL2 family proteins used in the PACE experiments were ~230 amino acids long, indicating that a mutation occurred on average every ~250 phage replications. Phage population sizes ranged from $10^5$ per mL to $10^{10}$ per mL over the course of a PACE experiment, indicating a rate of 400–40,000,000 new mutations every generation. Conservative estimates thus suggest that a during each individual replicate, phage populations sampled at least 40,000 mutations, and upwards of $4 \times 10^9$ mutations. While not all mutations were equally likely each generation because MP6 enriches for transitions (i.e. G→A, A→G, C→T, and T→C), the high number of mutations sampled suggests that the vast majority of possible single point mutations (approximately

230*3*4 = 2760 potential mutations) were sampled over the course of each experiment, with higher population sizes generating all potential single point mutations each generation.

## Plaque assays

Plaque assays were performed on 1059 *E. coli* cells (*Carlson et al., 2014*; *Hubbard et al., 2015*), which supply gene III (gIII) to phage in an activity-independent manner, to measure phage titers. Additionally, activity-dependent plaque assays were done on S1030 *E. coli* containing the desired accessory plasmids to determine the number of phage encoding a BCL-2 family protein with a given peptide-binding profile. All cells were grown to an $OD_{600}$ of approximately 0.6 during the day. Four serial dilutions were done in Eppendorf tubes by serially pipetting 1 µL of phage into 50 µL of cells to yield the following dilutions: 1/50, 1/2500, 1/125,000, and 1/6,250,000. 650 µL of top agar (0.7% agar with LB media) was added to each tube, which was then immediately spread onto a quad plate containing bottom agar (1.5% agar with LB media). Plates were incubated overnight at 37°C. Plaques were counted the following day, and plaque forming units (PFU) per mL was calculated using the following equation:

$$PFU = 1000 * A * 50^{4-B}$$

Where *A* is the number of plaques in a given quadrant, and *B* is the quadrant number where the phage were counted, in which one is the least dilute quadrant and four is the most dilute quadrant.

## Phage growth assays

Phage growth assays were performed by adding the following to a culture tube and shaking at 37°C for 6 hr: 1 mL of LB with the appropriate antibiotics (carbenicillin and kanamycin), 10 µL of saturated S1030 *E. coli* containing the accessory plasmids of interest, and ~1000 phage. Phage were then isolated by centrifugation at 13,000 rpm for 3 min, and PFU was determined by plaque assays using 1059 *E. coli* and the plaque assay protocol described above.

## High-throughput sequencing library construction

PACE samples were collected from each lagoon every 24 hr. The lagoon samples were centrifuged at 13,000 rpm for 3 min on a bench top centrifuge to separate supernatant and cell pellet. The phage-containing supernatants were stored at 4°C prior to the creation of sequencing libraries. To prepare Illumina sequencing libraries, each phage sample was cultured overnight with 1059 *E. coli* cells, followed by phage DNA purification (Qiagen plasmid purification reagent buffer), P1 (catalog number 19051), P2 (catalog number 19052), N3 (catalog number 19064), PE (catalog number 19065), and spin column for DNA (EconoSpin, catalog number 1920–250). The resulting DNA concentration was ~50 ng/µL. Freshly generated DNA samples were then used as template for PCR amplification. For each library sample, we amplified three overlapping fragments of the BCL-2 family protein, which are 218–241 bp in length (*Figure 4—figure supplement 1*). Each primer also included 6–9 'N's to introduce length variation (*Supplementary file 4*). In total, 12 PCRs were used for each library. Phusion DNA polymerases and buffers (ThermoFisher Scientific, catalog number F518L) were used in the first PCR round to amplify all three fragments for all library sequencing. The 25 µL reaction contained: 0.5 µL of 50 mM $MgCl_2$, 0.75 µL of 10 mM dNTP, 0.75 µL Phusion DNA polymerase, 20 ng library DNA, and 0.5 µL of 10 µM primer (each). The PCR were run on a C1000 Touch Thermal Cycler (Bio-Rad), with the following parameters: 98°C for 1 min, followed by 16 cycles of 98°C for 12 s, 58°C for 15 s and 72°C for 45 s, and finally 72°C for 5 min. PCR were purified using the ZYMO DNA clean and concentrator kit (catalog number D4013) and 96 well filter plate (EconoSpin, catalog number 2020–001). The DNA products were dissolved in 30 µL ddH$_2$O. All 12 reactions for each library were combined, and 1 µL was used as the template for a second PCR round. PCR components and thermocycler parameters were the same as above, except that the annealing temperature was 56°C, and only 15 rounds of amplification were conducted. The primer and sample combinations are listed in *Supplementary file 4*. PCRs were then purified following the same procedure as previous step. Equal volumes of all 72 library samples were combined and concentration was measured using a Qubit 4 Fluorometer. The total DNA sample was 2.68 ng/µL (equivalent to 10 nM, according to the average length of PCR fragments). DNA samples were diluted to 4 nM from step 4 following the Illumina MiSeq System Denature and Dilute Libraries Guide and then diluted to 12 pM for high-

throughput sequencing. The final sample contained 100 μL of 20 pM PhiX spike-in plus 500 μL of the 12 pM library sample. Sequencing was performed on the Illumina MiSeq System using MiSeq Reagent Kit v3 (600-cycle) with paired-end reads according to the manufacturer's instructions.

### Processing of Illumina data

Illumina sequencing yielded 22 million reads, 13 million of which could be matched to a specific sample (*Supplementary file 4*). One replicate for AncB5 was found to be contaminated and removed from further analysis. To process the remaining data, we first used Trim Galore with default settings to trim reads based on quality (https://www.bioinformatics.babraham.ac.uk/projects/trim_galore/). Then, we used BBMerge, a script in BBTools (https://jgi.doe.gov/data-and-tools/bbtools/), to merge paired-end reads. Next, we used Clumpify to remove repeated barcode sequences. We then used Seal to identify and bin reads by sample and fragment. Finally, we used BBDuk to remove any primer or adapter sequence present. Scripts and reference sequences are available on Github (*Thornton, 2021*).

### Illumina sequencing analysis

Reads were binned by experiment and then aligned to the appropriate WT sequence using Geneious (low sensitivity, five iterations, gaps allowed). Sequences were then processed in R to remove sequences containing 'N's or that were not full length. Insertions found in less than 1% of the population and sites that extended outside of the coding region were removed from all sequences. Remaining gaps were standardized among replicates and within an experiment. Finally, allele frequencies were calculated for each site and amino acid, as well as remaining insertions and deletions.

### Quantifying the effects of chance and contingency on the outcomes of evolution

#### Estimating the effects of chance

Allele frequency differences among replicates started from the same genotype can only be caused by chance events. Thus, to determine the effects of chance ($C_1$) on the outcomes of evolution, we compared allele frequencies from replicate PACE experiments started from the same genotype. We compared allele frequencies of individual replicates to the average allele frequencies among replicates started from the same genotype by estimating the probability that two randomly chosen alleles would be different, i.e. the genetic variance, for each replicate individually ($V_r$) and the pooled sample of all replicates from a given starting genotype ($V_g$):

$$C_1 \equiv \frac{V_g}{V_r} = \frac{1 - Q_g}{1 - Q_r}$$

where $Q_r$ is the probability that two randomly chosen alleles in the same replicate are identical in state and $Q_g$ is the probability that two randomly chosen alleles from the pooled replicates started from the same genotype are identical in state. $C_1$ is related to Wright's F$_{is}$ statistic as:

$$F_{is} \equiv F_{rg} \equiv \frac{Q_r - Q_g}{1 - Q_g}$$

$$1 - F_{rg} = \frac{1 - Q_g - Q_r + Q_g}{1 - Q_g} = \frac{1 - Q_r}{1 - Q_g}$$

$$\frac{1}{1 - F_{rg}} = \frac{1 - Q_g}{1 - Q_r} = C_1$$

We used count data from Illumina sequencing to estimate allele frequencies and followed the approach of *Hivert et al., 2018*, which developed a methods of moments estimator, $\hat{F}_{st}^{pool}$, that is appropriate for pooled data and accounts for both the sampling of individuals within a population and the sampling of reads during sequencing. We treated each amino acid site independently and defined the following:

$$R_{1:rgs} \equiv \text{\# of reads in replicate } r \text{ of starting genotype } g \text{ at site } s$$

$$\hat{\pi}_{a:rgs} \equiv \text{observed allele frequency of allele } a \text{ in replicate } r \text{ of starting genotype } g \text{ at site } s$$

$$\hat{\pi}_{a:gs} \equiv \text{observed allele frequency of allele } a \text{ in pooled replicates of starting genotype } g \text{ at site } s$$

Using these values, we used the estimator of $\hat{F}_{st}^{pool}$ defined in *Hivert et al., 2018* to estimate $F_{rg}$ for a single site:

$$\hat{F}_{st}^{pool} \equiv \hat{F}_{rg}^s \equiv \frac{MSG_{gs} - MSR_{gs}}{MSG_{gs} + (\Re_{gs} - 1)MSR_{gs}}$$

where:

$MSG_{gs} \equiv \frac{1}{\mathbb{R}_{1:gs} - \mathbb{R}_{2:gs}} \sum\limits_{a=1}^{21} \sum\limits_{r=1}^{n_{r:g}} R_{1:rgs}\left(\hat{\pi}_{a:rgs} - \hat{\pi}_{a:gs}\right)^2$ is the mean sum-of-squares for pooled replicates.

$MSR_{gs} \equiv \frac{1}{R_{1:gs} - \mathbb{R}_{1:gs}} \sum\limits_{a=1}^{21} \sum\limits_{r=1}^{n_{r:g}} R_{1:rgs}\hat{\pi}_{a:rgs}\left(1 - \hat{\pi}_{a:rgs}\right)$ is the mean sum-of-squares within replicates.

$\Re_{gs} \equiv \frac{R_{1:gs} - \frac{R_{2:gs}}{R_{1:gs}}}{\mathbb{R}_{1:gs} - \mathbb{R}_{2:gs}}$, is the effective number of individuals after accounting for sampling, with

$$R_{1:gs} \equiv \sum_{r=1}^{n_{r:g}} R_{1:rgs},$$

$$R_{2:gs} \equiv \sum_{r=1}^{n_{r:g}} R_{1:rgs}^2$$

$$\mathbb{R}_{1:gs} \equiv \sum_{r=1}^{n_{r:g}} \frac{R_{1:rgs} + n_{i:rg} - 1}{n_{i:rg}},$$

and

$$\mathbb{R}_{2:gs} \equiv \frac{1}{R_{1:gs}} \sum_{r=1}^{n_{r:g}} \frac{R_{1:rgs}\left(R_{1:rgs} + n_{i:rg} - 1\right)}{n_{i:rg}}.$$

Here, $n_{r:g}$ is the number of replicates started from genotype $g$ and $n_{i:rg}$ is the number of individual phage in replicate $r$ of starting genotype $g$ in the sample used to make the sequencing library.

From the relationship between $C_1$ and $F_{rg}$, we approximated the site-specific effects of chance for a particular starting genotype as:

$$\hat{C}_{1:g}^s \equiv \frac{1}{1 - \hat{F}_{rg}^s} = \frac{1}{1 - \frac{MSG_{gs} - MSR_{gs}}{MSG_{gs} + (\Re_{gs} - 1)MSR_{gs}}} = \frac{1}{\frac{MSG_{gs} + (\Re_{gs} - 1)MSR_{gs} - MSG_{gs} + MSR_{gs}}{MSG_{gs} + (\Re_{gs} - 1)MSR_{gs}}}$$

$$= \frac{MSG_{gs} + (\Re_{gs} - 1)MSR_{gs}}{\Re_{gs}MSR_{gs}} = 1 + \frac{MSG_{gs} - MSR_{gs}}{\Re_{gs}MSR_{gs}}$$

When there were replicates from more than one starting genotype, we calculated $MSG_{gs}$, $MSR_{gs}$, and $\Re_{gs}$ separately for each starting genotype and averaged these values together, using weights proportional to the number of replicates for that genotype. Thus:

$$\hat{C}_1^s = 1 + \frac{\sum\limits_{g=1}^{n_g} n_{r:g} \sum\limits_{g=1}^{n_g} n_{r:g}\left(MSG_{gs} - MSR_{gs}\right)}{\sum\limits_{g=1}^{n_g} n_{r:g}\Re_{gs} \sum\limits_{g=1}^{n_g} n_{r:g}MSR_{gs}}$$

where $n_g$ is the number of distinct starting genotypes.

We then took the average numerator and average denominator as suggested by *Hivert et al., 2018* and *Weir and Cockerham, 1984* for estimating $F_{st}$:

$$\hat{C}_1 = 1 + \frac{\sum\limits_{s=1}^{n_s} \sum\limits_{g=1}^{n_g} n_{r:g} \sum\limits_{g=1}^{n_g} n_{r:gs} \left( MSG_{gs} - MSR_{gs} \right)}{\sum\limits_{s=1}^{n_s} \sum\limits_{g=1}^{n_g} n_{r:g} \Re_{gs} \sum\limits_{g=1}^{n_g} n_{r:g} MSR_{gs}}$$

where $n_s$ is the number of sites.

## Estimating the effects of contingency

To determine the effects of contingency ($C_2$) on the outcomes of evolution, we compared the average allele frequency of replicate PACE experiments between different starting genotypes. For each starting genotype, we pooled all replicates started from that genotype and treated it as a single sample. We compared allele frequencies among genotypes by estimating the probability that two randomly chosen alleles in a sample would be different if they were both drawn from the same starting genotype $(V_g)$ or drawn from different starting genotypes $(V_t)$:

$$C_2 \equiv \frac{V_t}{V_g} \equiv \frac{1 - Q_{g1 \neq g2}}{1 - Q_{g1 = g2}}$$

where $Q_{g1=g2}$ is the probability that two randomly chosen alleles from the same starting genotype are identical in state and $Q_{g1 \neq g2}$ is the probability that two randomly chosen alleles from different starting genotypes are identical in state. To calculate $1 - Q_{g1 \neq g2}$, we note that the probability of two randomly drawn alleles being different when chosen irrespective of starting genotype is simply the average of the probability of two randomly drawn alleles being different when they are drawn from the same and different starting genotypes, that is:

$$(1 - Q_t) = \frac{1}{2} \left( 1 - Q_{g1=g2} \right) + \frac{1}{2} \left( 1 - Q_{g1 \neq g2} \right)$$

where $Q_t$ is the probability that two randomly chosen alleles irrespective of starting genotype are identical in state. From this we have:

$$2(1 - Q_t) = \left( 1 - Q_{g1=g2} \right) + \left( 1 - Q_{g1 \neq g2} \right)$$

$$\left( 1 - Q_{g1 \neq g2} \right) = 2(1 - Q_t) - \left( 1 - Q_{g1=g2} \right)$$

$$\left( 1 - Q_{g1 \neq g2} \right) = 1 - 2Q_t + Q_{g1=g2}$$

Using this and the fact that $Q_{g1=g2}$ is equivalent to $Q_g$ used above to calculate the effects of chance, we have:

$$C_2 = \frac{1 - 2Q_t + Q_{g1=g2}}{1 - Q_{g1=g2}} = \frac{1 - Q_t}{1 - Q_{g1=g2}} + \frac{Q_{g1=g2} - Q_t}{1 - Q_{g1=g2}} = \frac{1 - Q_t}{1 - Q_g} + \frac{Q_g - Q_t}{1 - Q_g}$$

This statistic is related to Wright's $F_{st}$ as:

$$F_{st} \equiv F_{gt} \equiv \frac{Q_g - Q_t}{1 - Q_t}$$

$$1 - F_{gt} = \frac{1 - Q_t - Q_g + Q_t}{1 - Q_t} = \frac{1 - Q_g}{1 - Q_t}$$

and

$$1 + F_{gt} = \frac{1 - Q_t}{1 - Q_t} + \frac{Q_g - Q_t}{1 - Q_t}$$

$$\frac{1+F_{gt}}{1-F_{gt}} = \frac{\frac{1-Q_t}{1-Q_t} + \frac{Q_g - Q_t}{1-Q_t}}{\frac{1-Q_g}{1-Q_t}} = \frac{1-Q_t}{1-Q_g} + \frac{Q_g - Q_t}{1-Q_g}$$

As with the effects of chance, we used the method of moments estimator defined in *Hivert et al., 2018* to estimate the effects of contingency:

$$\hat{F}_{gt} \equiv \frac{MST_s - MSG_s}{MST_s + (\Re_{ts} - 1)MSG_s}$$

where:

$MST_s \equiv \frac{1}{\mathbb{R}_{1:s} - \mathbb{R}_{2:s}} \sum\limits_{a=1}^{21} \sum\limits_{g=1}^{n_g} R_{1:gs}(\hat{\pi}_{a:gs} - \hat{\pi}_{a:s})^2$ is the mean sum-of-squares for the entire pooled sample.

$MSG_s \equiv \frac{1}{\mathbb{R}_{1:s} - \mathbb{R}_{1:s}} \sum\limits_{a=1}^{21} \sum\limits_{g=1}^{n_g} R_{1:gs}\hat{\pi}_{a:gs}(1 - \hat{\pi}_{a:gs})$ is the mean sum-of-squares for a starting genotype.

$\Re_{ts} \equiv \frac{R_{1:s} - \frac{R_{2:s}}{R_{1:s}}}{\mathbb{R}_{1:s} - \mathbb{R}_{2:s}}$, is the effective number of individuals after accounting for sampling, with

$\hat{\pi}_{a:s} \equiv$ the observed allele frequency of allele $a$ among all genotypes at site $s$,

$$R_{1:s} \equiv \sum_{g=1}^{n_g} R_{1:gs},$$

$$R_{2:s} \equiv \sum_{g=1}^{n_g} R_{1:gs}^2,$$

$$\mathbb{R}_{1:s} \equiv \sum_{g=1}^{n_g} \frac{R_{1:gs} + n_{i:g} - 1}{n_{i:g}},$$

and

$$\mathbb{R}_{2:s} \equiv \frac{1}{R_{1:s}} \sum_{g=1}^{n_g} \frac{R_{1:gs}(R_{1:gs} + n_{i:g} - 1)}{n_{i:g}}.$$

With $n_{i:g}$ being the number of individual phage used to make the libraries for starting genotype $g$. From the relationship between $F_{gt}$ and $C_2$, we approximated the effects of contingency as:

$$\hat{C}_2^s = \frac{1 + \hat{F}_{gt}^s}{1 - \hat{F}_{gt}^s} = \frac{1 + \frac{MSP_{ts} - MSI_{ts}}{MSP_{ts} + (\Re_{ts} - 1)MSI_{ts}}}{1 - \frac{MSP_{ts} - MSI_{ts}}{MSP_{ts} + (\Re_{ts} - 1)MSI_{ts}}}$$

$$= \frac{\frac{MSP_{ts} + (\Re_{ts} - 1)MSI_{ts} + MSP_{ts} - MSI_{ts}}{MSP_{ts} + (\Re_{ts} - 1)MSI_{ts}}}{\frac{MSP_{ts} + (\Re_{ts} - 1)MSI_{ts} - MSP_{ts} + MSI_{ts}}{MSP_{ts} + (\Re_{ts} - 1)MSI_{ts}}}$$

$$= \frac{MSP_{ts} + (\Re_{ts} - 1)MSI_{ts} + MSP_{ts} - MSI_{ts}}{MSP_{ts} + (\Re_{ts} - 1)MSI_t s - MSP_{ts} + MSI_{ts}}$$

$$= \frac{\Re_{ts}MSI_{ts} + 2MSP_{ts} - 2MSI_{ts}}{\Re_{ts}MSI_{ts}}$$

$$= 1 + 2\frac{MSP_{ts} - MSI_{ts}}{\Re_{ts}MSI_{ts}}$$

Again, we treated all sites as independent and summed the numerator and denominators to estimate the effects of contingency:

$$\hat{C}_2 = 1 + 2 \frac{\sum_{s=1}^{n_s}(MSP_{ts} - MSI_{ts})}{\sum_{s=1}^{n_s} \Re_{ts} MSI_{ts}}$$

## Estimating the combined effect of chance and contingency

To determine the combined effects of chance and contingency ($C_3$) on the outcomes of evolution, we compared allele frequencies from individual replicates to the average allele frequency among replicates from different starting genotypes. In each case, we pooled replicates started from a genotype and treated it as a single sample and compared it to the individual replicates started from different genotypes. We compared allele frequencies by estimating the probability that two randomly chosen alleles would be different if they were both drawn from the same replicate or if they were drawn from a different starting genotype:

$$C_3 \equiv \frac{1 - Q_{g1 \neq g2}}{1 - Q_r} = \frac{1 - Q_g}{1 - Q_r} * \frac{1 - Q_{g1 \neq g2}}{1 - Q_g} = C_1 * C_2 = \frac{V_t}{V_r}$$

We thus used:

$$\hat{C}_3 = \hat{C}_1 * \hat{C}_2$$

as our estimate of the combined effects of chance and contingency. This estimator indicates that the combined effects of chance and contingency are multiplicative and thus amplify each other's effects as they get larger.

# Acknowledgements

We thank members of the Thornton and Dickinson groups for helpful comments on the manuscript, S Ahmadiantehrani for editing, and R Ranganathan for the use of the Illumina MiSeq instrument.

# Additional information

## Competing interests

Jinyue Pu, Bryan C Dickinson: Has a patent on the proximity-dependent split RNAP technology used in this work (US Patent App. 16/305,298, 2020). The other authors declare that no competing interests exist.

## Funding

| Funder | Grant reference number | Author |
| --- | --- | --- |
| National Institutes of Health | R01GM131128 | Joseph W Thornton |
| National Institutes of Health | R01GM121931 | Joseph W Thornton |
| National Institutes of Health | R01GM139007 | Joseph W Thornton |
| National Institutes of Health | F32GM122251 | Brian PH Metzger |
| National Science Foundation | DGE-1746045 | Victoria Cochran Xie |
| National Science Foundation | 1749364 | Bryan C Dickinson |

The funders had no role in study design, data collection and interpretation, or the decision to submit the work for publication.

## Author contributions

Victoria Cochran Xie, Conceptualization, Investigation, Methodology, Writing - original draft, Writing - review and editing, Designed, engineered, optimized and implemented PACE dual-selection system. Performed PACE, biochemical assays, and sequencing experiments. Provided input on the phylogenetic, genetic, and evolutionary analyses; Jinyue Pu, Conceptualization, Investigation, Methodology, Writing - original draft, Writing - review and editing, Designed, engineered,

optimized and implemented the PACE dual-selection system. Performed PACE, biochemical assays, and sequencing experiments. Provided input on phylogenetic, genetic, and evolutionary analyses; Brian PH Metzger, Conceptualization, Data curation, Formal analysis, Funding acquisition, Methodology, Writing - original draft, Writing - review and editing, Provided input on the PACE, biochemical assays, and sequencing experiments. Developed and designed the evolutionary and genetic analyses. Led and performed the phylogenetic, genetic, and evolutionary analyses. Led writing and revision; Joseph W Thornton, Conceptualization, Supervision, Funding acquisition, Methodology, Writing - original draft, Project administration, Writing - review and editing, Developed and designed the evolutionary and genetic analyses. Led writing and revision; Bryan C Dickinson, Conceptualization, Resources, Supervision, Funding acquisition, Methodology, Writing - original draft, Project administration, Writing - review and editing, Designed the PACE dual-selection system

#### Author ORCIDs
Brian PH Metzger https://orcid.org/0000-0003-4878-2913
Joseph W Thornton https://orcid.org/0000-0001-9589-6994
Bryan C Dickinson http://orcid.org/0000-0002-9616-1911

#### Decision letter and Author response
Decision letter https://doi.org/10.7554/eLife.67336.sa1
Author response https://doi.org/10.7554/eLife.67336.sa2

## Additional files

### Supplementary files
• Supplementary file 1. Luciferase assay data for all experiments.

• Supplementary file 2. Posterior probabilities for reconstructed ancestral sequences. For each sequence, the site, maximum likelihood (ML) amino acid state, and posterior probability (PP) are given, along with the highest posterior probability alternative (ALT) state and posterior probability for this alternative state. Locations of paralog-specific insertions are shown as gaps. For each reconstructed sequence, the average posterior probability for the maximum likelihood states and the alternative states is given, as are the number of sites where the posterior probability of a non-maximum likelihood state is greater than 0.2. Finally, the average, maximum, minimum, and variance among reconstructed ancestors are given for the average maximum likelihood posterior probability and the number of non-maximum likelihood states greater than 0.2 posterior probability.

• Supplementary file 3. List of PACE experiments, amino acid alignments of hsBCL-2 and hsMCL-1 with their structural global alignment, and mutations found in individual variants isolated from PACE. fs is frameshift, aa is amino acid, co is codon change.

• Supplementary file 4. PACE library and high-throughput sequencing (HTS) data. PACE experiments are listed in the tab 'Library-info', which contains the name, purpose of the experiment, and HTS experiment numbers. The tab 'Primers for HTS' lists all the primer sequences used for HTS library constructions. The tab 'MiSeq reads number' include the read number of each library in this MiSeq run and the library sample information. The library samples are labeled as X*-end or X*-$$. 'X' indicates the specific PACE experiment, '*' the experimental replicate, 'end' means samples were collected after 96 hr when the experiment finished, and '$$' indicates the time point after removing chemostat A (e.g. 'B2-24' is a sample from replicate 2 of evolution B and collected 24 hr after removing chemostat A, which is 72 hr from the start of PACE). The tab 'genotype' includes the aligned protein sequences with corresponding residue numbers. The 'Frequency' tab contains the non-wild-type amino acid frequency of each sample for each site.

• Supplementary file 5. Descriptions of plasmids and sequences used.

• Transparent reporting form

## Data availability

The high throughput sequencing data of evolved BCL-2 family protein variants were deposited in the National Center for Biotechnology Information (NCBI) Sequence Read Archive (SRA) databases. They can be accessed via BioProject: PRJNA647218. The processed sequencing data are available on Dryad (https://doi.org/10.5061/dryad.866t1g1ns). The coding scripts and reference sequences for processing the data are available on Github (https://github.com/JoeThorntonLab/BCL2. ChanceAndContingency).

The following datasets were generated:

| Author(s) | Year | Dataset title | Dataset URL | Database and Identifier |
|---|---|---|---|---|
| Xie VC, Pu J, Metzger BPH, Thornton JW, Dickinson BC | 2020 | Experimental evolution of BCL2 family ancestral proteins | https://www.ncbi.nlm. nih.gov/bioproject/ PRJNA647218 | NCBI Bioproject, PRJNA647218 |
| Xie VC, Pu J, Metzger BPH, Thornton JW, Dickinson BC | 2020 | BCL2-Chance and Contingency | https://doi.org/10.5061/ dryad.866t1g1ns | Dryad Digital Repository, 10.5061/ dryad.866t1g1ns |

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
