## [Decision Letter]

**Acceptance summary:**

This manuscript presents an original, clever, high throughput, and rapid experimental protein evolution method to assess the roles and contributions of contingency, chance, and necessity in the evolution of protein-protein interactions. The authors focus on the animal BCL^-^2 protein family and on the evolution of their binding properties to two proteins, NOXA and BID. Using several replicates and several starting points, they found little predictability between replicates of single starting points and among those from multiple starting points, indicating that there is no single pathway through sequence space to the selected function, and that historical contingency is here the primary cause of protein evolution. The original experimental set up described in this paper allows to simultaneously impose selection and counter-selection within the same cell, and shows a lot of potential for future studies of directed protein evolution in general.

**Decision letter after peer review:**

Thank you for submitting your article "Contingency and chance erase necessity in the experimental evolution of ancestral proteins" for consideration by *eLife*. Your article has been reviewed by 3 peer reviewers, including Virginie Courtier-Orgogozo as the Reviewing Editor and Reviewer #1, and the evaluation has been overseen by Molly Przeworski as the Senior Editor. The following individual involved in review of your submission has agreed to reveal their identity: Zachary David Blount (Reviewer #3).

Essential revisions:

The experiments are very nice, very well done, clearly presented and well interpreted.

1) Our main essential revision is to ask the authors to tone down their conclusions and broad claims. Indeed, the setup only allowed the authors to study the evolution of one protein family, considering only protein-protein interactions with few players, and in an artificial, bacterial environment. In living organisms, each protein is likely to exhibit particular properties such that it can bind or not bind to hundreds of different proteins, and not just two as tested here. So the constraints present in living organisms may be much larger than the ones present within this experimental evolution set up. Furthermore, the tested proteins probably encounter other constraints in their native environment besides affinity for other proteins.

2) The authors should also acknowledge the limitations of their study: BID and NOXA ancestral proteins were not used, peptides instead of full-length protein were used during interaction assays, proteins partially membrane-bound in vivo.

*Reviewer #1 (Recommendations for the authors):*

For the reversion experiment, proteins are being evolved to gain NOXA binding but the experiment after 48h does not select for proteins that retain BID binding (Figure 3-supp2, E). So how do you explain that the evolved proteins did not loose their BID binding?

Paragraph "Contingency is the major cause of sequence variation on long timescales" (line 268). I am not entirely convinced by this paragraph. It would be good to repeat similar measurements in conditions where plasmids are left to evolve randomly, with no selection for particular novel protein binding properties. This would give an idea of the null hypothesis. Maybe similar numbers would also be found with such a "neutral" evolution.

Unless I am wrong, I would suggest to remove this part.

Line 239: Figure 4: how do you explain that there is quite a lot of variability in the phenotype for replicate 3 and for replicate 4 of the evolved hsMCL^-^1 in terms of NOXA binding (panel A). Did you check whether the 3 individual phenotypes shown for each replicate do have the same genotype? It looks like amino acid position 202 can be N or S for replicate 3. Could it explain the difference in binding between "individuals"?Can you also explain variability between "individual phenotypes" for the few replicates of AncM6 and AncB1?

*Reviewer #2 (Recommendations for the authors):*

The amount of experimental work reported here is already quite impressive, and additional experiments, although being useful to enforce the conclusions, may not be practically feasible. This should not be an issue if the authors are ready to tone down their interpretations. Instead of discussing immediate extension of the method to other families, they could indicate the limitations of their current experimental setup in order to indicate further directions for follow-up studies.

Figure 2—figure supplement 3: the authors do not discuss the lack of binding observed between Trichoplax adhaerens BCL and human BID and NOXA, and they also do not plot them on figure 2. Actually, it would be interesting to discuss this point in the light of the recent report by Popgeorgiev et al., (Science Advances, 2020, eabc4149) that the Trichoplax BCL^-^2L2 actually binds the Trichoplax Bak BH3 protein (called BCL^-^2L3).

*Reviewer #3 (Recommendations for the authors):*

To flesh out my comment about needing to engage the conceptual work that has been done on contingency, I have two major issues I would like to see dealt with:

First, contingency and chance are discussed by the authors as being separate, but they are not. Indeed, chance is a critical component of contingency. Some more engagement with the philosophers who have specialized in this area, especially John Beatty, Eric Desjardins and Alison McConwell. They have made clear that contingency is not just causal dependence. Beatty, for instance, has articulated how contingent outcomes are not simply those that stem from earlier events, but depend on pasts with chance components so that their occurrence was not guaranteed. (See Beatty, and Carrera, When what had to happen was not bound to happen: History, chance, narrative, evolution. Journal of the Philosophy of History 5: 471-495. 2011.) Unless chance is involved in the causal paths taken over time from a given start to a given beginning, then there is not really any contingency, but merely determinism. Desjardins has elaborated this further into understanding contingency in the context of path dependence (See Desjardins, Historicity and Experimental Evolution. Biol. Philos. 26, 339-364, 2011 And Desjardins, Reflections on path dependence and irreversibility: Lessons from evolutionary biology. Philosophy of Science 78: 724 – 738, 2017.) The passage in the introduction that, "…if diversity reflects contingency-a strong dependence of future events on past and current states-then the outcomes of evolution would be predictable only given complete knowledge of the constraints and opportunities specific to each starting point" is therefore incorrect, because contingency involves not just dependence on starting conditions. Contingency makes for broader unpredictability because indeterminacy, such as that introduced by the randomness of mutations, is characteristic of the causal chain no matter the starting point. (This point, incidentally, is made in the Blount et al., 2018 and Beatty, 2009 papers cited to support the quoted statement.) Similarly, the conclusion, then, that "Conversely, without chance, contingency in history would be inconsequential, because all phylogenetic lineages launched from a common ancestor would always lead to the same intermediate steps and thus the same ultimate outcomes" is well stated, but it misses that contingency without chance is not contingency. It misses, therefore, sophisticated conceptual elaboration of what exactly what contingency really involves. I think the arguments made about what the experimental findings mean would be better if that work were taken into account. The study of contingency is rife with conceptual mix-ups and crosstalk, and it is critical to engage what the philosophers have done to clarify things and help us scientists out so that we can really all be on the same page.

My second issue is related. The work deals with chance, contingency, and necessity in the evolution of protein sequence. However, there is little grappling with how contingency actually matters if it simply means that proteins can evolve along many different paths depending on historical substitutions to the same function. If there are multiple paths to functionally equivalent end states, how do they matter? Such convergence is in line with Conway Morris's argument that contingency does not matter because, "the routes are many, but the destinations are limited." Indeed, if there really are a fantastically large number of alternate paths that can be taken to the same end, then the role for contingency in the final state is actually minimal. Where it could matter would be if 1. There were end functional states reachable via different pathways that were qualitatively similar, but not actually the same, and/or 2. The end states vary in their potential for later evolution. As it stands, the manuscript leaves me unable to see how it matters if "Present-day proteins are physical anecdotes of the particular unpredictable histories", but all those anecdotes have the same punchlines and meanings. Now, I will state outright that I think that history does play a role beyond simply the contingency of evolutionary paths to the same destination, but there is a functionally convergent, non-contingent elephant in the room with which the authors need to engage.

---

## [Author Response]

Essential revisions:The experiments are very nice, very well done, clearly presented and well interpreted.1) Our main essential revision is to ask the authors to tone down their conclusions and broad claims. Indeed, the setup only allowed the authors to study the evolution of one protein family, considering only protein-protein interactions with few players, and in an artificial, bacterial environment. In living organisms, each protein is likely to exhibit particular properties such that it can bind or not bind to hundreds of different proteins, and not just two as tested here. So the constraints present in living organisms may be much larger than the ones present within this experimental evolution set up. Furthermore, the tested proteins probably encounter other constraints in their native environment besides affinity for other proteins.

We agree. We have revised the manuscript in numerous places to narrow our claims appropriately and acknowledge these limitations. Specifically:

– Throughout the abstract, results, and discussion, we have made our claims particular to the BCL^-^2 protein family. When we suggest generality to other proteins, we make the rationale for this extension explicit and label it as speculative (e.g. lines 30, 596, 640).

– We made explicit throughout the paper that we assessed chance and contingency in evolution under selection for new functions in the PACE laboratory evolution system, rather than during the natural historical evolution of BCL^-^2 family proteins. Because we used ancestral starting points, the contingency we observe is generated by historical sequence substitutions, but the outcomes that are contingent are within the PACE system (e.g. text beginning at lines 263 and 535).

– We expanded the portion of the discussion in which we discuss ways in which chance and contingency during history may be different from those in our experiments (line 582). We addressed the possible effects of other protein-protein interactions, the use of bacterial cells, and other functional constraints, as suggested.

2) The authors should also acknowledge the limitations of their study: BID and NOXA ancestral proteins were not used, peptides instead of full-length protein were used during interaction assays, proteins partially membrane-bound in vivo.

We agree and have modified the discussion to explicitly acknowledge each of the listed differences between our PACE system and the cellular context of natural BCL^-^2 family function (line 582). We discussed ways in which these differences could affect the roles of chance and contingency. Also, we narrowed our claims to pertain specifically to the effects of chance and contingency during laboratory evolution using PACE, as discussed above.

3) Reviewer 1 noted that Figure 4F shows that some PACE mutations recapitulated historical substitutions, suggesting that necessity may not have been almost entirely absent.

We understand and have clarified the argument in the Results section (lines 263-279). and revised the figures concerning the occurrence or reversion of historical substitutions in PACE trajectories. The PACE experiments show that virtually no mutations occurred in all trajectories from all starting points under either selection regime, indicating virtually no necessity under PACE conditions. The purpose of the analysis of historical substitutions was to gain insight into the extent of chance, contingency, and necessity during historical evolutionary change in BCL^-^2 PPI specificity. The key question, then, is whether substitutions that occurred on the phylogenetic branch when specificity was changed also occurred (or were reverted) during PACE selection for the derived (or ancestral) function, and if so, how frequently and in what backgrounds. This did not come through clearly in the original text and figure. We revised Figure 4F so that it now shows clearly that none of the substitutions from the key phylogenetic interval occurred in PACE; moreover, there were only two reversions, and these were observed in only a subset of replicates (indicating chance) and only in trajectories from the ancestral starting point closest to the branch on which they occurred historically (indicating contingency). This therefore indicates a lack of necessity in the historical evolution of BCL^-^2 specificity. Some substitutions from other branches did occur in some PACE trajectories, but these could not have been historical causes of the shift in specificity and therefore do not provide information about chance and necessity during that shift; those data are now in Figure 4—figure supplement 5.

4) Reviewer 1 noted that several PACE mutations may have had no effect on NOXA or BID biding and could therefore have occurred because of mutational bias, drift or hitchhiking. R1 said this means that one cannot include such changes when calculating the fraction of acquired states that are attributable to chance. R1 said that mutations that arose repeatedly during PACE replicates from any given starting genotype, and which do contribute to the change in specificity, provide evidence for necessity.

We addressed this comment in two ways. First, we note that there are two ways that chance can determine sequence changes that occur under selection for a new function: (a) mutations have no effect on function (and therefore occur by chance because of drift or hitchhiking, as the reviewer notes), or (b) they may confer a new function but may be one of several mutations (or sets of mutations) that can do so, with chance determining which mutation(s) are realized. In either of these cases, differences between trajectories launched from the same starting point are attributable to chance. We clarified the text to explicitly acknowledge this (lines 418-421).

Second, we clarified in the introduction and in Figure 1 that necessity and determinism are not the same thing (line 80). Necessity requires the absence of both chance and contingency: a process must be both deterministic and insensitive to starting/intervening conditions to give rise to necessary outcomes. The observation that the reviewer refers to – some mutations that occur repeatably in trajectories from the same starting point – indicates a limited degree of chance and partial determinism. However, these mutations did not occur from trajectories launched from different starting points, indicating a large role for contingency and therefore a lack of necessity.

5) Reviewer 1 noted that there may be more mutations that can confer loss of binding to another protein than mutations that confer gain of binding to a new one. They suggested that we discuss this point in more detail.”

This is an interesting question, and we have performed a new analysis to test this possibility. We compared the effects of chance and contingency for starting genotypes that gained or lost NOXA binding, finding no statistically significant difference between them (line 319). This analysis is shown in Figure 5—figure supplement 1. In addition, there was no qualitative difference in the ability of gains and loss of NOXA binding to evolve; in all cases the change in NOXA binding was readily acquired during PACE.

6) Reviewer 1 noted that proteins in their natural cellular context encounter many potential binding partners, whereas we selected only for binding and specificity to just two of the biological significant binding partners for BCL^-^2 family proteins. As a result, the constraints during natural evolution could be more stringent than those in our experiment, leading to an underestimate of the role of determinism.

This comment and our response is addressed in point #1 above.

7) Reviewer 1 asked us to explain why the evolved proteins that were selected to gain NOXA binding did not lose BID binding if they were not subject to continued selection for that function.

BCL^-^2 family proteins selected to gain NOXA binding were first subject to an initial acclimation period prior to NOXA selection in which they were selected to maintain BID binding. A likely explanation for why these population did not lose BID binding during selection for NOXA binding is that the acquired mutations that enhanced NOXA binding do not diminish BID binding. Supporting this view, we also found that when proteins that had acquired dual BID/NOXA activity in PACE were subsequently subjected to PACE with selection to lose their BID activity, NOXA-specific binders failed to evolve. This suggests strong coupling of NOXA binding with BID binding in sequence space accessible from these proteins. This conclusion is somewhat speculative, however, so we have decided not to include it in the paper.

8) Reviewer 1 commented about the paragraph "Contingency is the major cause of sequence variation on long timescales" that a useful reference experiment would be to perform PACE under conditions where plasmids are left to evolve randomly, with no selection for particular novel protein binding properties. This would give an idea of chance, contingency, and necessity under a null hypothesis of neutral evolution.

We agree that our experiments reveal the effects of historical substitutions on chance and contingency during selection for a new function, not under the neutral evolutionary scenario of purifying selection for an existing function. We have made this explicit in our description of our experiments and claims. In the discussion we note that additional experiments would be required to reveal the effects of chance and contingency under the neutral evolutionary scenario and reveal any differences from their effects under selection for a new function (line 593).

9) Reviewer 2 noted as an unnatural aspect of our design that we used extant human BID and NOXA and that during history these proteins would also have evolved.

We agree and have added this point to the discussion (line 579). Our experiments kept BID and NOXA constant, allowing us to estimate the effects of historical substitutions on chance and contingency during PACE experiments for altered binding to human BID and NOXA. In reality, BID and NOXA would have evolved, presenting more opportunities for chance and contingency to influence the outcomes of evolution, thus making our analysis a conservative test of these factors.

10) Reviewer 2 noted that cross-species interactions among BCL2 family proteins may produce different results than using proteins that existed in the same organism at the same time and cited (Popgeorgiev et al., Science Advances 2021) as an example.

The referenced paper provides relevant information, and we have now cited it in the manuscript (line 763). Specifically, experiments in that paper show that the Trichoplax BCL2 family protein most closely related to vertebrate BCL2 is capable of binding both human BID and NOXA. This is the expected result based on our ancestral reconstruction, which show that BCL^-^2 at the time of its origin by gene duplication could bind both coactivators, and that NOXA binding was lost after placozoa diverged from the lineage leading to other animals (cnidaria, protostomes, and deuterostomes).

11) Reviewer 2 noted that binding affinity in the BCL2 family is often dependent on the length of the protein or peptide used.

We have added this point explicitly to the discussion as a difference between our assay conditions and natural conditions (line 588). We have also noted that the data in the cited review shows that the length of the protein or peptide affects absolute affinity but does not alter specificity (relative affinity for coactivator proteins). In addition, we altered the introduction to indicate that human BCL^-^2 (unlike human MCL^-^1) strongly prefers BID to NOXA, although that preference is not absolute (line 93).

12) Reviewer 2 commented that BCL2 family proteins are often membrane-located, while our experiments use a soluble binding assay.

We have added this point explicitly to the discussion as a difference between our assay conditions and natural conditions (line 584). We also pointed out that interactions with BID and NOXA naturally occur in the cytosol for BCL2 family proteins and are mediated via a cytosol-exposed hydrophobic cleft even when the BCL2 family proteins are membrane-bound. Thus, although there are critical downstream dimensions of BCL^-^2 family protein functions not addressed in our experiments, our particular focus – coactivator specificity – can be reasonably, albeit cautiously, studied using a cytosolic assay system.

13) Reviewer 2 suggested that “Instead of discussing immediate extension of the method to other families, they could indicate the limitations of their current experimental setup in order to indicate further directions for follow-up studies.”

We followed this suggestion and added an extended discussion of these and other limitations of our experimental system and the implications of those limitations for our findings, before we bring up potential extension to other proteins (text starting at line 582).

14) Reviewer 3 noted that chance and contingency interact with one another and that several philosophers of science have shown that both chance and contingency are required for history to matter in the outcomes of evolution.

We agree and have adjusted the introduction and discussion accordingly, adding citations to Beatty and to Desjardins as suggested (e.g. text starting at lines 44 and 553). Our goal with these changes is to make clear that chance and contingency are conceptually distinct, but they strongly interact with each other. Chance–defined as random occurrence of one event from a probability distribution of multiple possibilities– is manifest as distinct outcomes among replicates from the same starting point under identical conditions. In contrast, contingency – differences in the probability distribution of events that can ensue from different starting points – is manifest as distinct outcomes among pooled replicates from different starting points. Under some circumstances, contingency can be realized or observed only if chance is also present: for example, in an evolutionary process beginning from a common ancestor, if chance is absent, each lineage would undergo the same intermediate steps and would therefore deterministically evolve the same states. Even if different outcomes would evolve from different starting points or intermediate states – that is, contingency still exists in the underlying structure of the system itself – those differences would never be realized or observed. If multiple starting points are used initially, however, contingency will be apparent in the outcomes even in the absence of chance. Conversely, chance can exist without contingency, but it would have no further effect on future paths or outcomes, and therefore would have no meaningful evolutionary consequences. We have modified the text, legend, and Figure 1A to make this thinking clearer.

15) Reviewer 3 commented that contingency is not simply conditionality on the starting point, but is instead dependence on intermediate steps and is thus more accurately referred to as path dependence.

We agree and have altered the text throughout to note that contingency is defined as differences in the probability of outcomes that depend on the starting point or on intervening events in evolutionary trajectories (e.g. lines 46, 118, 559). We also explicitly used the phrase path-dependence to evoke this concept.

16) Reviewer 3 commented that our focus on chance and contingency in the evolution of molecular sequences is less consequential or interesting than those phenomena would be at the level of phenotype.

Our view is that both sequences and phenotypes are important biological objects, and that the evolutionary causes of variation are of great interest at both levels. Sequences and the sequence-structure-function relationship are central objects of study in biochemistry and molecular biology; and the sequence-function-phenotype relationship is the central object in molecular genetics and genomics. In these fields, mining and interpreting patterns of sequence variation is a key activity to gain insight into these relationships, so understanding how and why these patterns are produced during evolution is necessary for accurate interpretation of those patterns – e.g., are differences between orthologs or paralogs attributable to selection imposed by differences in environment/function, to a lack of constraint that yields random variation, or to differences in internal constraints that have arisen among divergent lineages? This has profound implications for how we interpret that variation. Moreover, sequences have long been a central focus of inference in molecular evolution, with interpretation of variation patterns in sequences being used to ask questions like how much of the genome – and what parts of it – are the result of different forms of selection, drift, and so on. Chance, contingency, and necessity in sequence evolution is therefore of major interest in its own right.

We modified the text in three ways to address this issue. First, we have made clear throughout that our results reveal the role of these modes of causality in molecular sequences (e.g. lines 239, 262, 305, 515). Second, we explicitly argue in the introduction and discussion why understanding how they affect sequence evolution is important (lines 104, 632). Finally, some of our results also illuminate chance, contingency, and necessity at the level of the PPI phenotype itself, with an observation of some necessity (convergence under selection) but some contingency (inaccessibility of some phenotype from some starting points under selection), and we discuss this in the last section of the results and in the discussion (lines 489, 626).